# Class-Aware Adversarial Transformers for Medical Image Segmentation

**Chenyu You**[1]     **Ruihan Zhao**[2]     **Fenglin Liu**[3]     **Siyuan Dong**[1]     **Sandeep Chinchali**[2]

**Ufuk Topcu**[2]          **Lawrence Staib**[1]          **James S. Duncan**[1]

[1]Yale University     [2]UT Austin     [3]University of Oxford

## Abstract

Transformers have made remarkable progress towards modeling long-range dependencies within the medical image analysis domain. However, current transformer-based models suffer from several disadvantages: (1) existing methods fail to capture the important features of the images due to the naive tokenization scheme; (2) the models suffer from information loss because they only consider single-scale feature representations; and (3) the segmentation label maps generated by the models are not accurate enough without considering rich semantic contexts and anatomical textures. In this work, we present `CASTformer`, a novel type of adversarial transformers, for 2D medical image segmentation. First, we take advantage of the pyramid structure to construct multi-scale representations and handle multi-scale variations. We then design a novel class-aware transformer module to better learn the discriminative regions of objects with semantic structures. Lastly, we utilize an adversarial training strategy that boosts segmentation accuracy and correspondingly allows a transformer-based discriminator to capture high-level semantically correlated contents and low-level anatomical features. Our experiments demonstrate that `CASTformer` dramatically outperforms previous state-of-the-art transformer-based approaches on three benchmarks, obtaining 2.54%-5.88% absolute improvements in Dice over previous models. Further qualitative experiments provide a more detailed picture of the model's inner workings, shed light on the challenges in improved transparency, and demonstrate that transfer learning can greatly improve performance and reduce the size of medical image datasets in training, making `CASTformer` a strong starting point for downstream medical image analysis tasks.

## 1   Introduction

Accurate and consistent measurements of anatomical features and functional information on medical images can greatly assist radiologists in making accurate and reliable diagnoses, treatment planning, and post-treatment evaluation [1]. Convolutional neural networks (CNNs) have been the de-facto standard for medical image analysis tasks. However, such methods fail in explicitly modeling long-range dependencies due to the intrinsic locality and weight sharing of the receptive fields in convolution operations. Such a deficiency in context modeling at multiple scales often yields sub-optimal segmentation capability in capturing rich anatomical features of variable shapes and scales (*e.g.*, tumor regions with different structures and sizes) [2, 3]. Moreover, using transformers has been shown to be more promising in computer vision [4–9] for utilizing long-range dependencies than other, traditional CNN-based methods. In parallel, transformers with powerful global relation modeling abilities have become the standard starting point for training on a wide range of downstream

36th Conference on Neural Information Processing Systems (NeurIPS 2022).

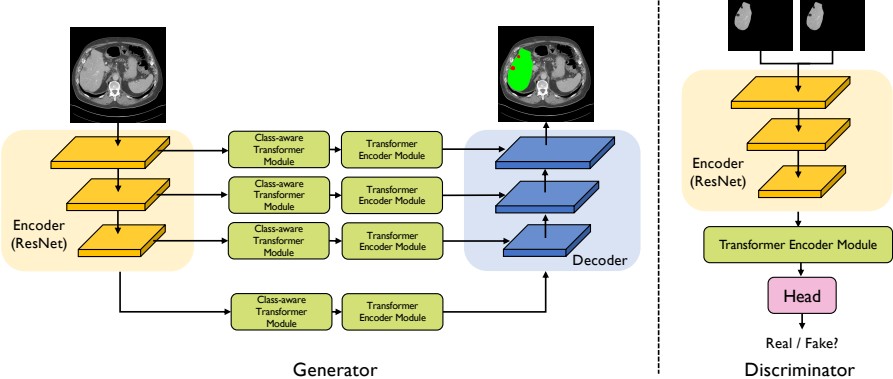

Figure 1: Our proposed `CASTformer` consists of a transformer-based generator (*i.e.*, `CATformer`) and a discriminator.

medical imaging analysis tasks, such as image segmentation [7, 10–13], image synthesis [14–16], and image enhancement [17–26].

Medical image semantic segmentation can be formulated as a typical dense prediction problem, which aims at performing pixel-level classification on the feature maps. Recently, Chen *et al.* [7] introduced `TransUNet`, which inherits the advantage of both `UNet` [27] and `Transformers` [4], to exploit high-resolution informative representations in the spatial dimension by CNNs and the powerful global relation modeling by Transformers. Although existing transformer-based approaches have proved promising in the medical image segmentation task, there remain several formidable challenges, because (1) the model outputs a single-scale and low-resolution feature representation; (2) prior work mainly adopts the standard tokenization scheme, hard splitting an image into a sequence of image patches of size $16 \times 16$, which may fail to capture inherent object structures and the fine-grained spatial details for the downstream dense prediction task; (3) compared to the standard convolution, the transformer architecture requires a grid structure, and thus lacks the capability to localize regions that contain objects of interest instead of the uninteresting background; and (4) existing methods are usually deficient in ensuring the performance without capturing both global and local contextual relations among pixels. We argue that transformer-based segmentation models are not yet robust enough to replace CNN-based methods, and investigate several above-mentioned key challenges transformer-based segmentation models still face.

Inspired by recent success of vision transformer networks [3, 4, 28–34], we make a step towards a **more practical scenario** in which we only assume access to pre-trained models on public computer vision datasets, and a relatively small medical dataset, which we can use the weights of the pre-trained models to achieve higher accuracy in the medical image analysis tasks. These settings are particularly appealing as (1) such models can be easily adopted on typical medical datasets; (2) such a setting only requires limited training data and annotations; and (3) transfer learning typically leads to better performance [35–38]. Inspired by such findings, we propose several novel strategies for expanding its learning abilities to our setting, considering both multi-scale anatomical feature representations of interesting objects and transfer learning in the medical imaging domain.

First, we aim to model multi-scale variations by learning feature maps of different resolutions. Thus, we propose to incorporate the pyramid structure into the transformer framework for medical image segmentation, which enables the model to capture rich global spatial information and local multi-scale context information. Additionally, we consider medical semantic segmentation as a sequence-to-sequence prediction task. The standard patch tokenization scheme in [4] is an art—splitting it into several fixed-size patches and linearly embedding them into input tokens. Even if significant progress is achieved, model performance is likely to be sub-optimal. We address this issue by introducing a novel class-aware transformer module, drawing inspiration from a progressive sampling strategy in image classification [39], to adaptively and selectively learn interesting parts of objects. This essentially allows us to obtain effective anatomical features from spatial attended regions within the medical images, so as to guide the segmentation of objects or entities.

Second, we adopt the idea of Generative Adversarial Networks (GANs) to improve segmentation performance and correspondingly enable a transformer-based discriminator to learn low-level anatom-

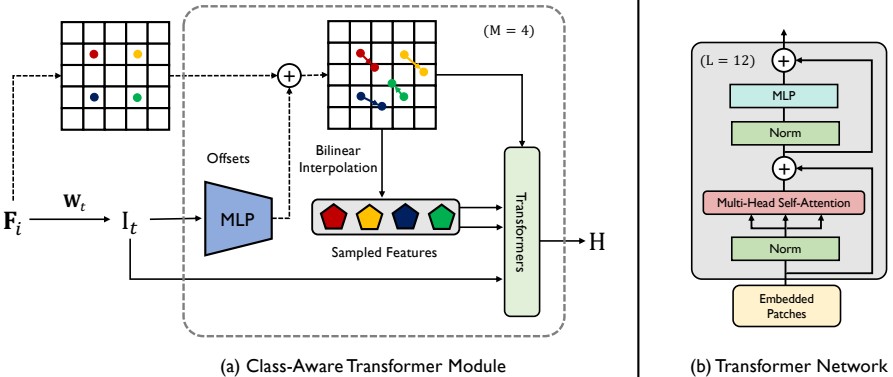

| (a) Class-Aware Transformer Module | (b) Transformer Network |

Figure 2: An illustration of (a) our class-aware transformer module and (b) transformer network. Our class-aware transformer module iteratively samples discriminative locations. A group of points is initialized in the regularly-spaced location. At each step, given the feature map $\mathbf{F}_i$, we iteratively update its sampling locations by adding them with the estimated offsets of the last step. Note that only 4 points are shown for a clear presentation, and there are more points in the actual implementation.

ical features and high-level semantics. The standard GANs are not guaranteed to prioritize the most informative demonstrations on interesting anatomical regions, and mixing irrelevant regions (*i.e.*, background) creates inferior contexts, which drastically underperform segmentation performance [2, 40, 3]. Additionally, it is well-known that they are notoriously difficult to train and prone to model collapse [41]. Training vision transformers is also tedious and requires large amounts of labeled data, which largely limits the training quality. We use a more refined strategy, where, for each input, we combine it with the predicted segmentation mask to create the image with anatomical demonstrations. We also leverage the pre-trained checkpoints to compensate the need of large-dataset training, thereby providing a good starting point with more discriminative visual demonstrations. We refer to our approach as CASTformer, **c**lass-**a**ware adver**s**arial **t**rans**former**s: a strong transformer-based method for 2D medical image segmentation. Our contributions are summarized as follows:

- **Novel Network Architecture:** We make the first attempt to build a GAN using a transformer-based architecture for the 2D medical image segmentation task. We incorporate the pyramid structure into the generator to learn rich global and local multi-scale spatial representations, and also devise a novel class-aware transformer module by progressively learning the interesting regions correlated with semantic structures of images. To the best of our knowledge, we are the **first work** to explore these techniques in the context of medical imaging segmentation.

- **Better Understanding Inner Workings:** We conduct careful analyses to understand the model's inner workings, how the sampling strategy works, and how different training factors lead to the final performance. We highlight that it is more effective to progressively learn distinct contextual representations with the class-aware transformer module, resulting in more accurate and robust models that applied better to a variety of downstream medical image analysis tasks (See Appendix **??** and **??**).

- **Remarkable Performance Improvements:** CASTformer contributes towards a dramatic improvement across three datasets we evaluate on. For instance, we achieve Dice scores of $82.55\%$ and $73.82\%$ by obtaining gains up to $5.88\%$ absolute improvement compared to prior methods on the Synapse multi-organ dataset. We illustrate the benefits of leveraging pre-trained models from the computer vision domain, and provide suggestions for future research that could be less susceptible to the confounding effects of training data from the natural image domain.

## 2  Related Work

**CNN-based Segmentation Networks**  Before the emergence of transformer-based methods, CNNs were the *de facto* methods in medical image segmentation tasks [42–61]. For example, Ronneberger *et al.* [27] proposed a deep 2D UNet architecture, combining skip connections between

opposing convolution and deconvolution layers to achieve promising performance on a diverse set of medical segmentation tasks. Han *et al.* [62] developed a 2.5D 24-layer Fully Convolutional Network (FCN) for liver segmentation tasks where the residual block was incorporated into the model. To further improve segmentation accuracy, Kamnitsas *et al.* [63] proposed a dual pathway 11-layer 3D CNN, and also employed a 3D fully connected conditional random field (CRF) [64] as an additional pairwise constraint between neighboring pixels for the challenging task of brain lesion segmentation.

**Transformers in Medical Image Segmentation**   Recent studies [7, 10, 12, 13, 65–70] have focused on developing transformer-based methods for medical image analysis tasks. Recently, Chen *et al.* [7] proposed `TransUNet`, which takes advantage of both `UNet` and `Transformers`, to exploit high-resolution informative information in the spatial dimension by CNNs and the global dependencies by Transformers. Cao *et al.* [10] explored how to use a pure transformer for medical image analysis tasks. However, the results do not lead to better performance. These works mainly utilized hard splitting some highly semantically correlated regions without capturing the inherent object structures. In this work, beyond simply using the naive tokenization scheme in [4, 7], we aim at enabling the transformer to capture global information flow to estimate offsets towards regions of interest.

**Transformer in Generative Adversarial Networks**   Adversarial learning has proved to be a very useful and widely applicable technique for learning generative models of arbitrarily complex data distributions in the medical domain. As the discriminator $D$ differentiates between real and fake samples, the adversarial loss serves as the regularization constraint to enforce the generator $G$ to predict more realistic samples. Inspired by such recent success [71–74, 22, 75, 20, 23, 21, 76, 77], Jiang *et al.* [71] proposed to build a GAN pipeline with two pure transformer-based architectures in synthesizing high-resolution images. Esser *et al.* [31] first used a convolutional GAN model to learn a codebook of context-rich visual features, followed by transformer architecture to learn the compositional parts. Hudson *et al.* [3] proposed a bipartite self-attention on StyleGAN to propagate latent variables to the evolving visual features. Despite such success, it requires high computation costs due to the quadratic complexity, which fundamentally hinders its applicability to the real world. Besides the image generation task, we seek to take a step forward in tackling the challenging task of 2D medical image segmentation.

## 3   Method

Our proposed approach is presented in Figure 1. Given the input image $\mathbf{x} \in \mathbb{R}^{H \times W \times 3}$, similar to `TransUNet` architecture [7], our proposed generator network $G$, termed `CATformer`, is comprised of four key components: encoder (feature extractor) module, class-aware transformer module, transformer encoder module, and decoder module. As shown in Figure 1, our generator has four stages with four parallel subnetworks. All stages share a similar architecture, which contains a patch embedding layer, class-aware layer, and $L_i$ Transformer encoder layers.

**Encoder Module.** Our method adopts a CNN-Transformer hybrid model design instead of using a pure transformer, which uses 40 convolutional layers, to generate multi-scale feature maps. Such a *convolutional stem* setting provides two advantages: (1) using *convolutional stem* helps transformers perform better in the downstream vision tasks [78–84]; (2) it provides high-resolution feature maps with parallel medium- and low-resolution feature maps to help boost better representations. In this way, we can construct the feature pyramid for the Transformers, and utilize the multi-scale feature maps for the downstream medical segmentation task. With the aid of feature maps of different resolutions, our model is capable of modeling multi-resolution spatially local contexts.

**Hierarchical Feature Representation.** Inspired by recent success in object detection [85], we deviate from `TransUNet` by generating a single-resolution feature map, and our focus is on extracting CNN-like multi-level features $\mathbf{F}_i$, where $i \in \{1, 2, 3, 4\}$, to achieve high segmentation accuracy by leveraging high-resolution features and low-resolution features. More precisely, in the first stage, we utilize the encoder module to obtain the dense feature map $\mathbf{F}_1 \in \mathbb{R}^{\frac{H}{2} \times \frac{W}{2} \times C_1}$, where $(\frac{H}{2}, \frac{W}{2}, C_1)$ is the spatial feature resolution and the number of feature channels. In a similar way, we can formulate the following feature maps as follows: $\mathbf{F}_2 \in \mathbb{R}^{\frac{H}{2} \times \frac{W}{2} \times (C_1 \cdot 4)}$, $\mathbf{F}_3 \in \mathbb{R}^{\frac{H}{4} \times \frac{W}{4} \times (C_1 \cdot 8)}$, and $\mathbf{F}_4 \in \mathbb{R}^{\frac{H}{8} \times \frac{W}{8} \times (C_1 \cdot 12)}$. Then, we divide $\mathbf{F}_1$ into $\frac{HW}{16^2}$ patches with the patch size $P$ of $16 \times 16 \times 3$, and feed the flattened patches into a learnable linear transformation to obtain the patch embeddings of size $\frac{HW}{16^2} \times C_1$.

**Class-Aware Transformer Module.** The class-aware transformer module (CAT) is designed to adaptively focus on useful regions of objects (*e.g.*, the underlying anatomical features and structural information). Our CAT module is largely inspired by the recent success for image classification [39], but we deviate from theirs as follows: (1) we remove the vision transformer module in [39] to alleviate the computation and memory usage; (2) we use $4$ separate Transformer Encoder Modules (TEM), which will be introduced below; (3) we incorporate $M$ CAT modules on multi-scale representations to allow for contextual information of anatomical features to propagate into the representations. Our class-aware transformer module is an iterative optimization process. In particular, we apply the class-aware transformer module to obtain the sequence of tokens $\mathbf{I}_{M,1} \in \mathbb{R}^{C \times (n \times n)}$, where $(n \times n)$ and $M$ are the number of samples on each feature map and the total iterative number, respectively. As shown in Figure 2, given the feature map $\mathbf{F}_1$, we iteratively update its sampling locations by adding them with the estimated offset vectors of the last step, which can be formulated as follows:

$$\mathbf{s}_{t+1} = \mathbf{s}_t + \mathbf{o}_t, \ t \in \{1, \ldots, M-1\}, \tag{1}$$

where $\mathbf{s}_t \in \mathbb{R}^{2 \times (n \times n)}$, and $\mathbf{o}_t \in \mathbb{R}^{2 \times (n \times n)}$ are the sampling location and the predicted offset vector at $t$-th step. Specifically, the $\mathbf{s}_1$ is initialized at the regularly spaced sampling grid. The $i$-th sampling location $\mathbf{s}_1^i$ is defined as follows:

$$\mathbf{s}_1^i = [\beta_i^y \tau_h + \tau_h/2, \beta_i^x \tau_w + \tau_w/2], \tag{2}$$

where $\beta_i^y = \lfloor i/n \rfloor$, $\beta_i^x = i - \beta_i^y * n$. The step sizes in the $y$ (row index) and $x$ (column index) directions denote $\tau_h = H/n$ and $\tau_w = W/n$, respectively. $\lfloor \cdot \rfloor$ is the floor operation. We can define the initial token on the input feature map in the following form: $\mathbf{I}_t' = \mathbf{F}_i(\mathbf{s}_t)$, where $t \in \{1, \ldots, M\}$, and $\mathbf{I}_t' \in \mathbb{R}^{C \times (n \times n)}$ denotes the initial sampled tokens at $t$-th step. We set the sampling function as the bilinear interpolation, since it is differentiable with respect to both the sampling locations $\mathbf{s}_t$ and the input feature map $\mathbf{F}_i$. We do an element-wise addition of the current positional embedding of the sampling locations, the initial sampled tokens, and the estimated tokens of the last step, and then we can obtain the output tokens at each step:

$$\begin{aligned}
\mathbf{S}_t &= \mathbf{W}_t \mathbf{s}_t \\
\mathbf{V}_t &= \mathbf{I}_t' \oplus \mathbf{S}_t \oplus \mathbf{I}_{t-1} \\
\mathbf{I}_t &= \text{Transformer}(\mathbf{V}_t), \ t \in \{1, \ldots, M\},
\end{aligned} \tag{3}$$

where $\mathbf{W}_t \in \mathbb{R}^{C \times 2}$ is the learnable matrix that embeds $\mathbf{s}_t$ to positional embedding $\mathbf{S}_t \in \mathbb{R}^{C \times (n \times n)}$, and $\oplus$ is the element-wise addition. Transformer$(\cdot)$ is the transformer encoder layer, as we will show in the following paragraphs. We can compute the estimated sampling location offsets as:

$$\mathbf{o}_t = \theta_t(\mathbf{I}_t), \ t \in \{1, \ldots, M-1\}, \tag{4}$$

where $\theta_t(\cdot) \in \mathbb{R}^{2 \times (n \times n)}$ is the learnable linear mapping for the estimated sampling offset vectors. It is worth noting that these operations are all differentiable, thus the model can be learned in an end-to-end fashion.

**Transformer Encoder Module.** The transformer encoder module (TEM) is designed to model long-range contextual information by aggregating global contextual information from the complete sequences of input image patches embedding. In implementations, the transformer encoder module follows the architecture in ViT [4], which is composed of Multi-head Self-Attention (MSA), and MLP blocks, which can be formulated as:

$$\mathbf{E}_0 = [\mathbf{x}_p^1 \mathbf{H}; \ \mathbf{x}_p^2 \mathbf{H}; \cdots; \ \mathbf{x}_p^N \mathbf{H}] + \mathbf{H}_{pos}, \tag{5}$$

$$\mathbf{E}'_i = \text{MSA}(\text{LN}(\mathbf{E}_{i-1})) + \mathbf{E}_{i-1}, \tag{6}$$

$$\mathbf{E}_i = \text{MLP}(\text{LN}(\mathbf{E}'_i)) + \mathbf{E}'_i, \tag{7}$$

where $i = 1 \ldots M$, and $\text{LN}(\cdot)$ is the layer normalization. $\mathbf{H} \in \mathbb{R}^{(P^2 \cdot C) \times D}$ and $\mathbf{H}_{pos} \in \mathbb{R}^{N \times D}$ denote the patch embedding projection and the position embedding.

**Decoder Module.** The decoder is designed to generate the segmentation mask based on four output feature maps of different resolutions. In implementations, rather than designing a hand-crafted decoder module that requires high computational demand, we incorporate a lightweight All-MLP decoder [86], and such a simple design allows us to yield a powerful representation much more

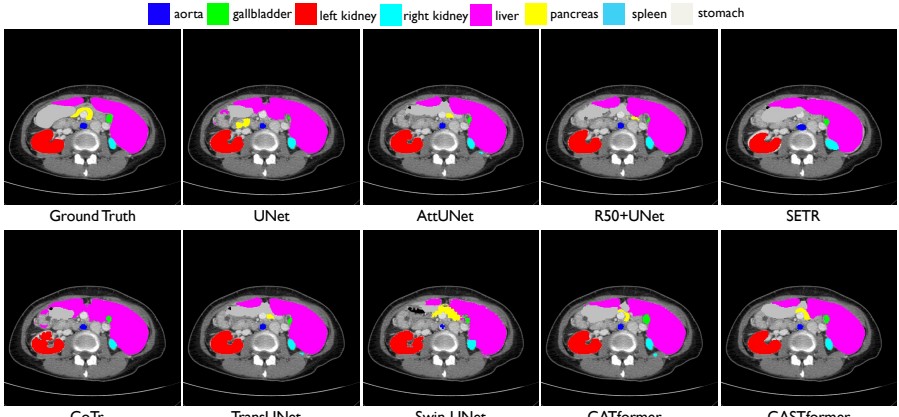

Figure 3: Visual comparisons with other methods on Synapse dataset. As observed, `CASTformer` achieves superior performance with detailed anatomical features and the boundary information of different organs.

efficiently. The decoder includes the following criteria: 1) the channel dimension of multi-scale features is unified through the MLP layers; 2) we up-sample the features to 1/4th and concatenate them together; 3) we utilize the MLP layer to fuse the concatenated features, and then predict the multi-class segmentation mask $\mathbf{y}'$ from the fused features.

**Discriminator Network.** We use the R50+ViT-B/16 hybrid model pre-trained on ImageNet from ViT [4] as a starting point for our discriminator design, in this case using the pre-trained strategies to learn effectively on the limited size target task data. Then, we simply apply a two-layer multi-layered perceptron (MLP) to make a prediction about the identity of the class-aware image. Following previous work [2], we first utilize the ground truth image $\mathbf{x}$ and the predicted segmentation mask $\mathbf{y}'$ to obtain the class-aware image $\tilde{\mathbf{x}}$ (*i.e.*, pixel-wise multiplication of $\mathbf{x}$ and $\mathbf{y}'$). It is important to note that this construction re-uses the pre-trained weights and does not introduce any additional parameters. $D$ seeks to classify between real and fake samples [87]. $G$ and $D$ compete with each other through attempting to reach an equilibrium point of the minimax game. Using this structure enables the discriminator to model long-range dependencies, making it better assess the medical image fidelity. This also essentially endows the model with a more holistic understanding of the anatomical visual modality (categorical features).

**Training Objective.** As to the loss function and training configurations, we adopt the settings used in Wasserstein GAN (WGAN) [88], and use WGAN-GP loss [89]. We jointly use the segmentation loss [7, 13] and WGAN-GP loss to train $G$. Concretely, the segmentation loss includes the dice loss and cross-entropy loss. Hence, the training process of `CASTformer` can be formulated as:

$$\mathcal{L}_G = \lambda_1 \mathcal{L}_{\text{CE}} + \lambda_2 \mathcal{L}_{\text{DICE}} + \lambda_3 \mathcal{L}_{\text{WGAN-GP}}, \tag{8}$$

where $\lambda_1, \lambda_2, \lambda_3$ determine the importance of each loss term. See Appendix **??** for an ablation study.

## 4 Experimental Setup

**Datasets.** We experiment on multiple challenging benchmark datasets: Synapse[1], LiTS, and MP-MRI. More details can be found in Appendix **??**.

**Implementation Details.** We utilize the AdamW optimizer [90] in all our experiments. For training our generator and discriminator, we use a learning rate of $5e^{-4}$ with a batch size of 6, and train each model for 300 epochs for all datasets. We set the sampling number $n$ on each feature map and the total iterative number $M$ as 16 and 4, respectively. See Appendix **??**, **??**, **??** for details on the training configuration, model architecture and hyperparameters. We also adopt the input resolution and patch size $P$ as 224×224 and 14, respectively. We set $\lambda_1 = 0.5$, $\lambda_2 = 0.5$, and $\lambda_3 = 0.1$ in this experiments. In the testing stage, we adopt four metrics to evaluate the segmentation performance: Dice coefficient (Dice), Jaccard Index (Jaccard), 95% Hausdorff Distance (95HD), and Average

---

[1] `https://www.synapse.org/#!Synapse:syn3193805/wiki/217789`

Table 1: Quantitative segmentation results on the Synapse multi-organ CT dataset.

| Encoder | Decoder | DSC ↑ | Jaccard ↑ | 95HD ↓ | ASD ↓ | Aorta | Gallbladder | Kidney (L) | Kidney (R) | Liver | Pancreas | Spleen | Stomach |
|---|---|---|---|---|---|---|---|---|---|---|---|---|---|
| | UNet [27] | 70.11 | 59.39 | 44.69 | 14.41 | 84.00 | 56.70 | 72.41 | 62.64 | 86.98 | 48.73 | 81.48 | 67.96 |
| | AttnUNet [91] | 71.70 | 61.38 | 34.47 | 10.00 | 82.61 | 61.94 | 76.07 | 70.42 | 87.54 | 46.70 | 80.67 | 67.66 |
| ResNet50 | UNet [27] | 73.51 | 63.81 | 29.65 | 8.83 | 82.21 | 55.06 | 76.71 | 73.07 | 89.36 | 53.52 | 84.91 | 73.22 |
| ResNet50 | AttnUNet [91] | 74.74 | 62.69 | 33.04 | 9.49 | 83.68 | 58.63 | 79.08 | 74.53 | 90.81 | 55.76 | 83.80 | 71.68 |
| | SETR [92] | 66.30 | 54.19 | 29.09 | 7.16 | 66.63 | 38.34 | 74.45 | 68.49 | 92.18 | 35.91 | 83.01 | 71.41 |
| | CoTr w/o CNN-encoder [13] | 54.82 | 42.49 | 69.58 | 20.37 | 63.22 | 37.86 | 67.10 | 60.61 | 88.48 | 15.46 | 60.74 | 45.12 |
| | CoTr [13] | 72.60 | 61.25 | 41.55 | 12.42 | 83.27 | 60.41 | 79.58 | 73.01 | 91.93 | 45.07 | 82.84 | 64.67 |
| | TransUNet [7] | 77.48 | 64.78 | 31.69 | 8.46 | 87.23 | 63.13 | 81.87 | 77.02 | 94.08 | 55.86 | 85.08 | 75.62 |
| | SwinUNet [10] | 76.33 | 65.64 | 27.16 | 8.32 | 85.47 | 66.53 | 83.28 | 79.61 | 94.29 | 56.58 | 90.66 | 76.60 |
| | • CATformer (ours) | 82.17 | 73.22 | **16.20** | **4.28** | 88.98 | 67.16 | 85.72 | 81.69 | 95.34 | 66.53 | 90.74 | 81.20 |
| | ◦ CASTformer (ours) | **82.55** | **74.69** | 22.73 | 5.81 | 89.05 | 67.48 | 86.05 | 82.17 | 95.61 | 67.49 | 91.00 | 81.55 |

Symmetric Surface Distance (ASD). All our experiments are implemented in Pytorch 1.7.0. We train all models on a single NVIDIA GeForce RTX 3090 GPU with 24GB of memory.

## 5 Results

We compare our approaches (*i.e.*, CATformer and CASTformer) with previous state-of-the-art transformer-based segmentation methods, including UNet [27], AttnUNet [91], ResNet50+UNet [27], ResNet50+AttnUNet [91], SETR [92], CoTr w/o CNN-encoder [13], CoTr [13], TransUNet [7], SwinUnet [10] on Synapse, LiTS, and MP-MRI datasets. More results are in Appendix **??** and **??**.

**Experiments: Synapse Multi-organ.** The quantitative results on the Synapse dataset are shown in Table 1. The results are visualized in Figure 3. It can be observed that our CATformer outperforms the previous best model by a large margin and achieves a $4.69\% - 8.44\%$ absolute improvement in Dice and Jaccard, respectively. Our CASTformer achieves the best performance of $82.55\%$ and $74.69\%$, dramatically improving the previous state-of-the-art model (TransUNet) by $+5.07\%$ and $+9.91\%$, in terms of both Dice and Jaccard scores. This shows that the anatomical visual information is useful for the model to gain finer control in localizing local semantic regions. As also shown in Table 1, our CASTformer achieves absolute Dice improvements of $+2.77\%$, $+2.51\%$, $+1.35\%$, $+4.95\%$ on large organs (*i.e.*, left kidney, right kidney, liver, stomach) respectively. Such improvements demonstrate the effectiveness of learning the evolving anatomical features of the image, as well as accurately identifying the boundary information of large organs. We also observed similar trends that, compared to the previous state-of-the-art results, our CASTformer obtains $89.05\%$, $67.48\%$, $67.49\%$ in terms of Dice on small organs (*i.e.*, aorta, gallbladder, pancreas) respectively, which yields big improvements of $+1.82\%$, $+0.95\%$, $+10.91\%$. This clearly demonstrates the superiority of our models, allowing for a spatially finer control over the segmentation process.

**Experiments: LiTS.** To further evaluate the effectiveness of our proposed approaches, we compare our models on the LiTS dataset. Experimental results on the LiTS CT dataset are summarized in Appendix Table **??**. As is shown, we observe that our CATformer yields a $72.39\%$ Dice score, outperforming all other methods. Moreover, our CASTformer significantly outperforms all previous approaches, including the previous best TransUNet, and establishes a new state-of-the-art of $73.82\%$ and $64.91\%$ in terms of Dice and Jaccard, which are $5.88\%$ and $4.66\%$ absolute improvements better than TransUNet. For example, our CASTformer achieves the best performance of $95.88\%$ Dice on the liver region by $2.48\%$, while it dramatically increases the result from $42.49\%$ to $51.76\%$ on the tumor region, demonstrating that our model achieves competitive performance on liver and tumor segmentation. As shown in Figure 4, our method is capable of predicting high-quality object segmentation, considering the fact that the improvement in such a setting is challenging. This demonstrates: (1) the necessity of adaptively focusing on the region of interests; and (2) the efficacy of semantically correlated information. Compared to previously high-performing models, our two approaches achieve significant improvements on all datasets, demonstrating their effectiveness.

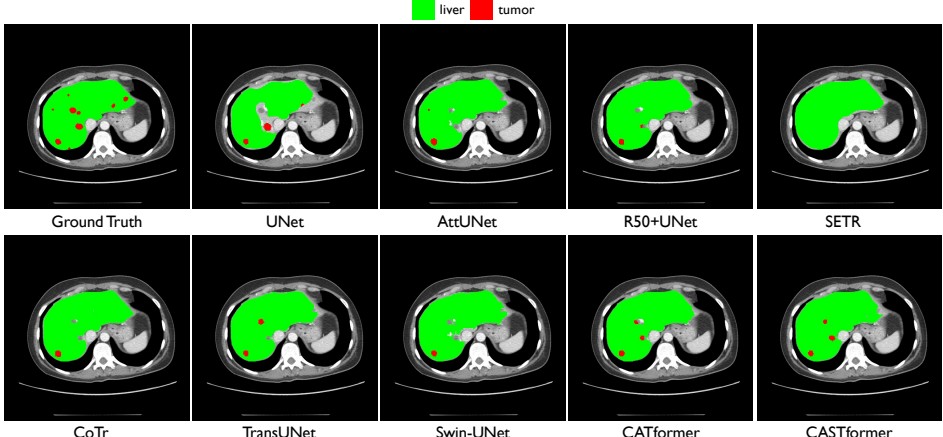

Figure 4: Visual comparisons with other methods on LiTS dataset. As observed, `CASTformer` achieves superior performance with detailed anatomical information (*e.g.*, the tumor regions in red).

## 6 Analysis

In this section, we conduct thorough analyses of our `CASTformer` in the following aspects: transfer learning, model components, effects of iteration number $N$ (Appendix **??**), sampling number $n$ (Appendix **??**), hyperparameter selection (Appendix **??**), different GAN-based loss functions (Appendix **??**), and gain a better understanding of the model's inner workings (Appendix **??** and **??**).

**Transfer Learning.** We consider whether we can leverage the pre-trained model commonly used in computer vision literature [4], to provide more evidence for the beneficial impact of our network performance. We use `CATformer` ($G$) as the baseline and evaluate all the settings on the Synapse multi-organ dataset. To put our results in perspective, we compare with six ways of using pre-trained R50+ViT-B/16 hybrid model from ViT [4] for transfer learning, namely (1) `CATformer`: w/o pre-trained and w/ pre-trained; (2) `CASTformer`: *both* w/o pre-trained, *only* w/ pre-trained $D$, *only* w/ pre-trained $G$, and *both* w/ pre-trained $G$ and $D$.

Table 2: Effect of transfer learning in our `CATformer` and `CASTformer` on the Synapse multi-organ dataset.

| Model | DSC | Jaccard | 95HD | ASD |
|---|---|---|---|---|
| ● CATformer (w/o pre-trained) | 74.84 | 65.61 | 31.81 | 7.23 |
| ● CATformer (w/ pre-trained) | 82.17 | 73.22 | 16.20 | 4.28 |
| ○ CASTformer (*both* w/o pre-trained) | 73.64 | 62.68 | 42.77 | 11.76 |
| ○ CASTformer (*only* w/ pre-trained $D$) | 78.87 | 69.36 | 30.54 | 9.17 |
| ○ CASTformer (*only* w/ pre-trained $G$) | 81.46 | 71.80 | 27.36 | 6.91 |
| ○ CASTformer (*both* w/ pre-trained) | 82.55 | 74.69 | 22.73 | 5.81 |

Table 3: Ablation on model component: Baseline; `CATformer` w/o CAT; `CATformer` w/o TEM; and `CATformer`.

| Model | DSC | Jaccard | 95HD | ASD |
|---|---|---|---|---|
| Baseline | 77.48 | 64.78 | 31.69 | 8.46 |
| ● CATformer w/o CAT | 80.09 | 70.56 | 25.62 | 7.30 |
| ● CATformer w/o TEM | 81.35 | 72.66 | 24.43 | 7.17 |
| ● CATformer | 82.17 | 73.22 | 16.20 | 4.28 |
| ○ CASTformer | 82.55 | 74.69 | 22.73 | 5.81 |

The results are in Table 2. Overall, we observe that using "w/ pre-trained" leads to higher accuracy than "w/o pre-trained", with significant improvements for the smaller sizes of datasets, suggesting that using "w/ pre-trained" provides us a good set of initial parameters for the downstream tasks. With using pre-trained weights, `CATformer` outperforms the setting without using pre-trained weights by a large margin and achieves $7.33\%$ and $7.61\%$ absolute improvements in terms of Dice and Jaccard. `CASTformer` ("both w/ pre-trained") also yields big improvements ($+8.91\%$ and $+12.01\%$) in Dice and Jaccard. This suggests that `CASTformer` is better at both initializing from the pre-trained models and better at gathering the anatomical information in a more adaptive and selective manner. As shown in Table 2, surprisingly, there is a significant discrepancy between only using "w/ pre-trained $D$" and "w/ pre-trained $G$": for example, `CASTformer` achieves $78.87\%$ in Dice with only w/ pre-trained $D$, while `CASTformer` achieves $81.46\%$ if only $G$ uses the pre-trained weights. This demonstrates that only using pre-trained weight in $D$ might be the culprit for the exploitation of anatomical information.

Our results suggest that (1) utilizing pre-trained models in the computer vision domain can help the model quickly adapt to new downstream medical segmentation tasks *without* re-building billions of anatomical representations; (2) we find that leveraging pre-trained weights can further boost the

performance because it can mitigate the discrepancy between training and inference; and (3) it also creates a possibility to adapt our model to the typical medical dataset with the smaller size.

**Ablation of Model Components.** Our key observation is that it is crucial to build high-quality anatomical representations through each model component. To show the strengths of our approach, we examine the following variants and inspect each key component on the Synapse multi-organ segmentation dataset: (1) **Baseline**: we remove the class-aware transformer module and the transformer encoder module in our `CATformer` as the baseline, similar to `TransUNet` defined in [7]; (2) `CATformer` **w/o CAT**: we *only* remove the class-aware transformer module in our `CATformer`; (3) `CATformer` **w/o TEM**: we *only* remove the transformer encoder module in our `CATformer`; (4) `CATformer`: this is our $G$ model; and (5) `CASTformer`: this is our final model described in Section 3. Table 3 summarizes the results of all the variants. As is shown, we observe that compared to the baseline model, both `CATformer` w/o CAT and `CATformer` w/o TEM are able to develop a better holistic understanding of global shapes/structures and fine anatomical details, thus leading to large performance improvements ($+2.61\%$ and $+3.87\%$) in terms of Dice. Our results show the class-aware transformer module is useful in improving the segmentation performance, suggesting that the discriminative regions of medical images are particularly effective. Finally, one thing worth noticing is that incorporating both the class-aware transformer module and the transformer encoder module performs better than *only* using a single module, highlighting the importance of two modules in our `CATformer`.

# 7 Conclusion and Discussion of Broader Impact

In this work, we have introduced `CASTformer`, a simple yet effective type of adversarial transformers, for 2D medical image segmentation. The key insight is to integrate the multi-scale pyramid structure to capture rich global spatial information and local multi-scale context information. Furthermore, `CASTformer` also benefits from our proposed class-aware transformer module to progressively and selectively learn interesting parts of the objects. Lastly, the generator-discriminator design is used to boost segmentation performance and correspondingly enable the transformer-based discriminator to capture low-level anatomical features and high-level semantics. Comprehensive experiments demonstrate that our `CASTformer` outperforms the previous state-of-the-art on three popular medical datasets considerably. We conduct extensive analyses to study the robustness of our approach, and form a more detailed understanding of desirable properties in the medical domain (*i.e.*, transparency and data efficiency).

Overall, we hope that this model can serve as a solid baseline for 2D medical image segmentation and motivate further research in medical image analysis tasks. It also provides a new perspective on transfer learning in medical domain, and initially shed novel insights towards understanding neural network behavior. As such pattern is hard to quantify, we expect more mechanistic explanations for clinical practise. We also plan to optimize the transformer-based architectures for the downstream medical image analysis tasks both in terms of data and model parameters.

**Broader Impact.** We acknowledge that our research will not pose significant risks of societal harm to society. Our work is scientific nature and will have the potential to positively contribute to a number of real-world clinical applications that establish high-quality and end-to-end medical image segmentation systems. We expect our approach to contribute to the grand goal of building more secured and trustworthy clinical AI.

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
