# Class-Aware Adversarial Transformers for Medical Image Segmentation

**Chenyu You**[1]    **Ruihan Zhao**[2]    **Fenglin Liu**[3]    **Siyuan Dong**[1]    **Sandeep Chinchali**[2]

**Ufuk Topcu**[2]    **Lawrence Staib**[1]    **James S. Duncan**[1]

[1]Yale University    [2]UT Austin    [3]University of Oxford

## A    Datasets

**Synapse**: Synapse multi-organ segmentation dataset includes 30 abdominal CT scans with 3779 axial contrast-enhanced abdominal clinical CT images. Each CT volume consists of $85 \sim 198$ slices of $512 \times 512$ pixels, with a voxel spatial resolution of $([0.54 \sim 0.54] \times [0.98 \sim 0.98] \times [2.5 \sim 5.0]) \text{mm}^3$. The dataset is randomly divided into 18 volumes for training (2212 axial slices), and 12 for validation. For each case, 8 anatomical structures are aorta, gallbladder, spleen, left kidney, right kidney, liver, pancreas, spleen, stomach.

**LiTS**: MICCAI 2017 Liver Tumor Segmentation Challenge (LiTS) includes 131 contrast-enhanced 3D abdominal CT volumes for training and testing. The dataset is assembled by different scanners and protocols from seven hospitals and research institutions. The image resolution ranges from 0.56mm to 1.0mm in axial and 0.45mm to 6.0mm in z direction. The dataset is randomly divided into 100 volumes for training, and 31 for testing.

**MP-MRI**: Multi-phasic MRI dataset is an in-house dataset including multi-phasic MRI scans of 20 local patients with HCC, each of which consisted of T1 weighted DCE-MRI images at three-time points (pre-contrast, arterial phase, and venous phases). Three images are mutually registered to the arterial phase images, with an isotropic voxel size of 1.00 mm. The dataset is randomly divided into 48 volumes for training, and 12 for testing.

## B    More Implementation Details

The training configuration and hyperparameter settings are summarized in Table 1.

## C    Model Architecture

We present the detailed architecture of `CATformer`'s encoding pipeline in Section 2. We use input/output names to indicate the direction of the data stream. `CATformer` applies independent class-aware attention on 4 levels of features extracted by the `ResNetV2` model. Each feature level L-$k$ is processed by `CATformer`-$k$, consisting of 4 blocks of class-aware transformer modules, followed by 12 layers of transformer encoder modules. Outputs from all four feature levels are fed into the decoder pipeline to generate the segmentation masks.

## D    More Experiments: LiTS

Experimental results are summarized in Table 3.

36th Conference on Neural Information Processing Systems (NeurIPS 2022).

Table 1: Training configuration and hyperparameter settings.

| Training Config | Hyperparameter |
|---|---|
| Optimizer | AdamW |
| Base learning rate | 5e-4 |
| Weight decay | 0.05 |
| Optimizer momentum | $\beta_1, \beta_2 = 0.9, 0.999$ |
| Batch size | 6 |
| Training epochs | 300 |
| Learning rate schedule | cosine decay |
| Warmup epochs | 5 |
| Warmup schedule | linear |
| Randaugment [1] | (9, 0.5) |
| Label smoothing [2] | 0.1 |
| Mixup [3] | 0.8 |
| Cutmix [4] | 1.0 |
| Gradient clip | None |
| Exp. mov. avg. (EMA) [5] | None |

Table 2: Architecture configuration of `CATformer`

| | | | CATformer | | |
|---|---|---|---|---|---|
| Stage | Layer | Input Name | Input Shape | Output Name | Output Shape |
| Encoder | ResNetV2 | Original Image | $224 \times 224 \times 3$ | RN-L1 RN-L2 RN-L3 RN-L4 | $112 \times 112 \times 64$ $56 \times 56 \times 256$ $28 \times 28 \times 512$ $14 \times 14 \times 1024$ |
| CATformer-1 | CAT$\times 4$ TEM$\times 12$ | RN-L1 CAT-1 | $112 \times 112 \times 64$ $(28 \times 28) \times 64$ | CAT-1 F1 | $(28 \times 28) \times 64$ $(28 \times 28) \times 64$ |
| CATformer-2 | CAT$\times 4$ TEM$\times 12$ | RN-L2 CAT-2 | $56 \times 56 \times 256$ $(28 \times 28) \times 256$ | CAT-2 F2 | $(28 \times 28) \times 256$ $(28 \times 28) \times 256$ |
| CATformer-3 | CAT$\times 4$ TEM$\times 12$ | RN-L3 CAT-3 | $28 \times 28 \times 512$ $(28 \times 28) \times 512$ | CAT-3 F3 | $(28 \times 28) \times 512$ $(28 \times 28) \times 512$ |
| CATformer-4 | CAT$\times 4$ TEM$\times 12$ | RN-L4 CAT-4 | $14 \times 14 \times 768$ $(14 \times 14) \times 768$ | CAT-4 F4 | $(14 \times 14) \times 768$ $(14 \times 14) \times 768$ |

# E   More Experiments: MP-MRI

Experimental results are summarized in Table 4. Overall, `CATformer` and `CASTformer` outperform the previous results in terms of Dice and Jaccard. Compared to SETR, our `CATformer` and `CASTformer` perform $1.78\%$ and $2.54\%$ higher in Dice, respectively. We also find `CASTformer` performs better than `CATformer`, which suggests that using discriminator can make the model better assess the medical image fidelity. Figure 1 shows qualitative results, where our `CATformer` and `CASTformer` provide better anatomical details than all other methods. This clearly demonstrates the superiority of our models. All these experiments are conducted using the same hyperparameters in our `CASTformer`.

# F   Effect of Iteration Number $N$

We explore the effect of different iteration number $N$ in Figure 2 (a). Note that in the case of $N = 1$, the sampling locations will not be updated. We find that more iterations of sampling clearly improve network performance in Dice and Jaccard. However, we observe that the network performance does not further increase from $N = 4$ to $N = 6$. In our study, we use $N = 4$ for the class-aware transformer module.

Table 3: Quantitative segmentation results on the LiTS dataset.

| Framework | | Average | | | | Liver | Tumor |
|---|---|---|---|---|---|---|---|
| Encoder | Decoder | DSC ↑ | Jaccard ↑ | 95HD ↓ | ASD ↓ | | |
| | UNet [6] | 62.88 | 54.64 | 57.59 | 27.74 | 88.27 | 37.49 |
| | AttnUNet [7] | 66.03 | 58.49 | 31.34 | 16.15 | 92.26 | 39.81 |
| ResNet50 | UNet [6] | 65.25 | 58.09 | 27.97 | 10.02 | 93.78 | 36.73 |
| ResNet50 | AttnUNet [7] | 66.22 | 59.27 | 31.47 | 10.41 | 93.26 | 39.18 |
| | SETR [8] | 54.79 | 49.21 | 36.34 | 15.04 | 91.69 | 17.90 |
| | CoTr w/o CNN-encoder [9] | 53.35 | 47.11 | 55.82 | 22.99 | 85.25 | 21.45 |
| | CoTr [9] | 62.67 | 55.43 | 34.75 | 15.84 | 89.43 | 35.92 |
| | TransUNet [10] | 67.94 | 60.25 | 29.32 | 12.45 | 93.40 | 42.49 |
| | SwinUNet [11] | 65.53 | 57.84 | 36.45 | 16.52 | 92.15 | 38.92 |
| | ● CATformer (ours) | 72.39 | 62.76 | **22.38** | 11.57 | 94.18 | 49.60 |
| | ○ CASTformer (ours) | **73.82** | **64.91** | 23.35 | **10.16** | 95.88 | 51.76 |

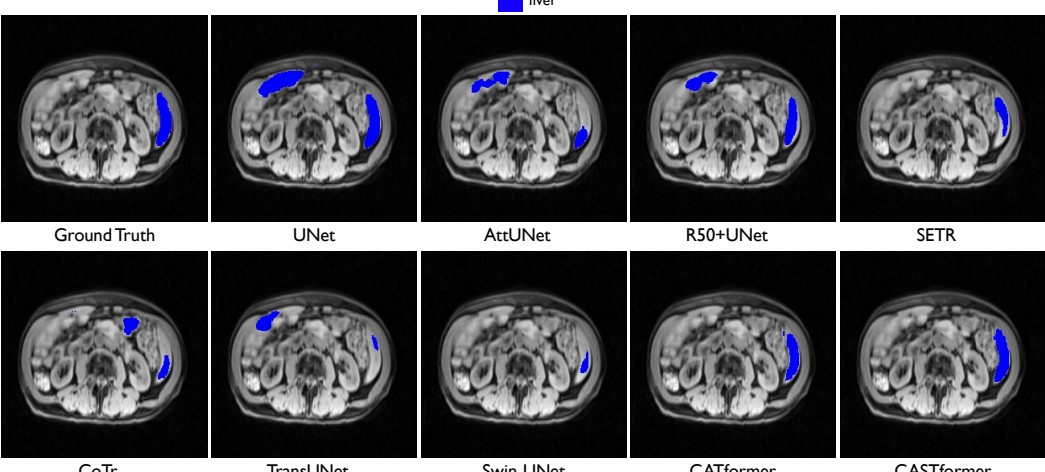

Figure 1: Visual comparisons with other methods on MP-MRI dataset.

# G    Effect of Sampling Number $n$

We further evaluate the effect of sampling number $n$ of the class-aware transformer module in Figure 2 (b). Empirically, we observe that results are generally well correlated when we gradually increase the size of $n$. As is shown, the network performance is optimal when $n = 16$.

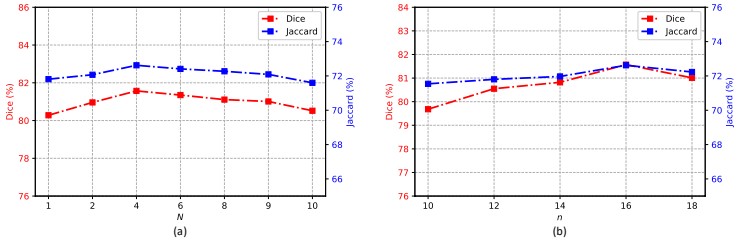

Figure 2: Effects of the iteration number $N$ and the sampling number $n$ in the class-aware transformer module. We report Dice and Jarrcd of CATformer on the Synapse multi-organ dataset.

# H    Hyperparameter Selection

We carry out grid-search of $\lambda_1, \lambda_2, \lambda_3 \in \{0.0, 0.1, 0.2, 0.5, 1.0\}$. As shown in Figure 3, with a carefully tuned hyperparameters $\lambda_1 = 0.5$, $\lambda_2 = 0.5$, and $\lambda_3 = 0.1$, such setting performs generally better than others.

Table 4: Quantitative segmentation results on the MP-MRI dataset.

| Framework | | Average | | | |
| Encoder | Decoder | DSC ↑ | Jaccard ↑ | 95HD ↓ | ASD ↓ |
| --- | --- | --- | --- | --- | --- |
| | UNet [6] | 88.38 | 79.42 | 39.23 | 11.14 |
| | AttnUNet [7] | 89.79 | 81.51 | 30.13 | 7.85 |
| ResNet50 | UNet [6] | 91.51 | 84.39 | 15.38 | 4.53 |
| ResNet50 | AttnUNet [7] | 91.43 | 84.24 | 14.14 | 4.24 |
| | SETR [8] | 92.39 | 85.89 | 7.66 | 3.79 |
| CoTr w/o CNN-encoder [9] | | 85.21 | 74.49 | 44.25 | 12.58 |
| | CoTr [9] | 90.06 | 81.94 | 28.91 | 7.89 |
| | TransUNet [10] | 92.08 | 85.36 | 23.17 | 6.03 |
| | SwinUNet [11] | 92.07 | 85.32 | 7.62 | 3.88 |
| ● CATformer (ours) | | 94.17 | 86.50 | **6.55** | 3.33 |
| ○ CASTformer (ours) | | **94.93** | **87.81** | 8.29 | **3.02** |

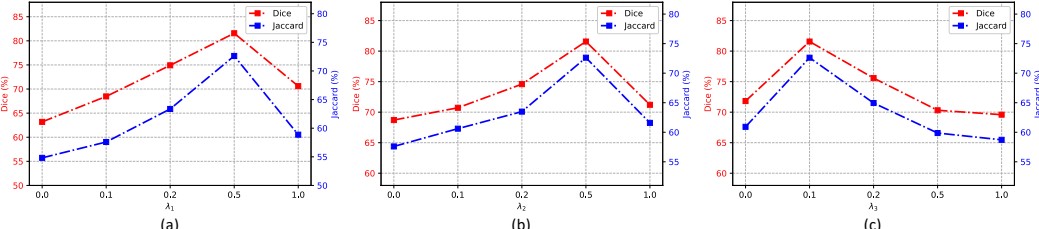

(a)  (b)  (c)

Figure 3: Effects of hyperparameters $\lambda_1, \lambda_2, \lambda_3$. We report Dice and Jarrcd of `CASTformer` on the Synapse multi-organ dataset.

# I   Importance of Loss Functions

One main argument for the discriminator is that modeling long-range dependencies and acquiring a more holistic understanding of the anatomical visual information can contribute to the improved capability of the generator. Besides the WGAN-GP loss [12], the minimax (MM) GAN loss [13], the Non-Saturating (NS) GAN loss [14], and Least Squares (LS) GAN Loss [15] are also commonly used as adversarial training. We test these alternatives and find that, in most cases, using WGAN-GP loss achieves comparable or higher performance than other loss functions. In addition, models trained using MM-GAN loss perform comparably to those trained using LS-GAN loss. In particular, our approach outperforms the second-best LS-GAN loss [15] by 1.10 and 2.49 points in Dice and Jaccard scores on the Synapse multi-organ dataset. It demonstrates the effectiveness of the WGAN-GP loss in our `CASTformer`.

Table 5: Ablation on Loss Function: MM-GAN loss [13]; NS-GAN loss [14]; LS-GAN loss [15]; and WGAN-GP loss [12].

| Model | DSC | Jaccard | 95HD | ASD |
| --- | --- | --- | --- | --- |
| MM-GAN loss [13] | 81.19 | 71.76 | 20.75 | 5.90 |
| NS-GAN loss [14] | 80.02 | 70.47 | 26.06 | 6.96 |
| LS-GAN loss [15] | 81.45 | 72.20 | 20.39 | 6.49 |
| WGAN-GP loss [12] | 82.55 | 74.69 | 22.73 | 5.81 |

# J   Visualization of Learned Sampling Location

To gain more insight into the evolving sampling locations learned by our proposed class-aware transformer module, we visualize the predicted offsets in Figure 4. We can see that particular sampling points around objects tend to attend to coherent segmented regions in terms of anatomical similarity and proximity. As is shown, we show the classes with the highly semantically correlated regions, indicating that the model coherently attends to anatomical concepts such as liver, right/left kidney, and spleen. These visualizations also illustrate how it behaves adaptively and distinctively to focus on the content with highly semantically correlated discriminative regions (*i.e.*, different organs). These findings can thereby suggest that our design can aid the `CATformer` to exercise finer

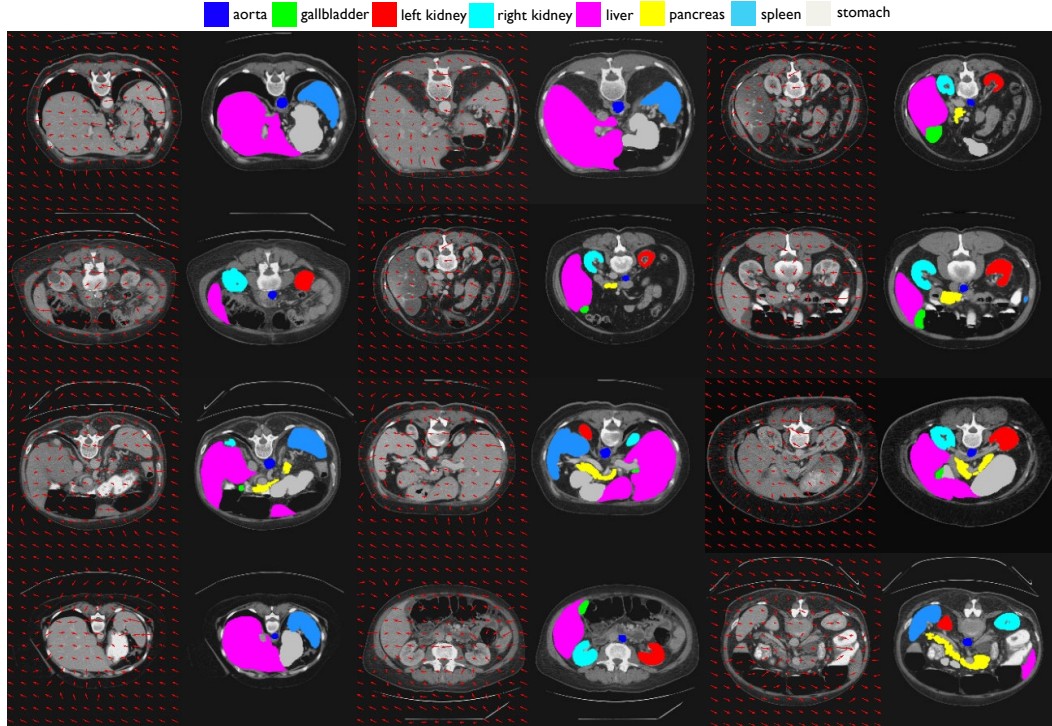

Figure 4: Visualization of sampled locations in the proposed class-aware transformer module.

control emphasizing anatomical features with the intrinsic structure at the object granularity. As is indicated (Figure 4 last column), we also find evidence that our model is prone to capture some small object cases (*e.g.*, pancreas, aorta, gallbladder). We hypothesize that it is because they contain more anatomical variances, which makes the model more difficult to exploit.

## K  Vision Transformer Visualization

In this section, we visualize the first 12 class-aware transformer layers on sequences of $28 \times 28$ feature patches in the encoder pipeline. In Figure 5, we plot the attention probabilities from a single patch over different layers and heads. Each row corresponds to one CAT layer; each column corresponds to an attention head. As we go deeper into the network, we are able to observe three kinds of attention behaviors as further discussed below.

**Attend to similar features:** In the first group of layers (layer 1 through 4), the attention probability is spread across a relatively large group of patches. Notably, these patches correspond to areas in the image with similar color and texture to the query patch. These more primitive attention distributions indicate that the class-awareness property has not yet been established.

**Attend to the same class and its boundary:** In the middle layers of the transformer model, most noticeable in the $5^{th}$ and $6^{th}$ layers, the attention probabilities start to concentrate on areas that share the same class label as the query patch (layer 5-2). In some other instances, the model attends to the boundary of the current class (layer 5-3, 5-6).

**Attend to other classes:** In the deeper layers of the model, the attention probability mainly concentrates on other classes. This clearly demonstrates persuasive evidence that the model establishes class awareness, which is helpful in the downstream medical segmentation tasks.

## L  More Ablations on Decoder Modules

In this section, we explore another state-of-the-art backbone proposed by Lin *et al.* [16], termed Feature Pyramid Network (FPN). FPN utilizes a top-down pyramid with lateral connections to construct

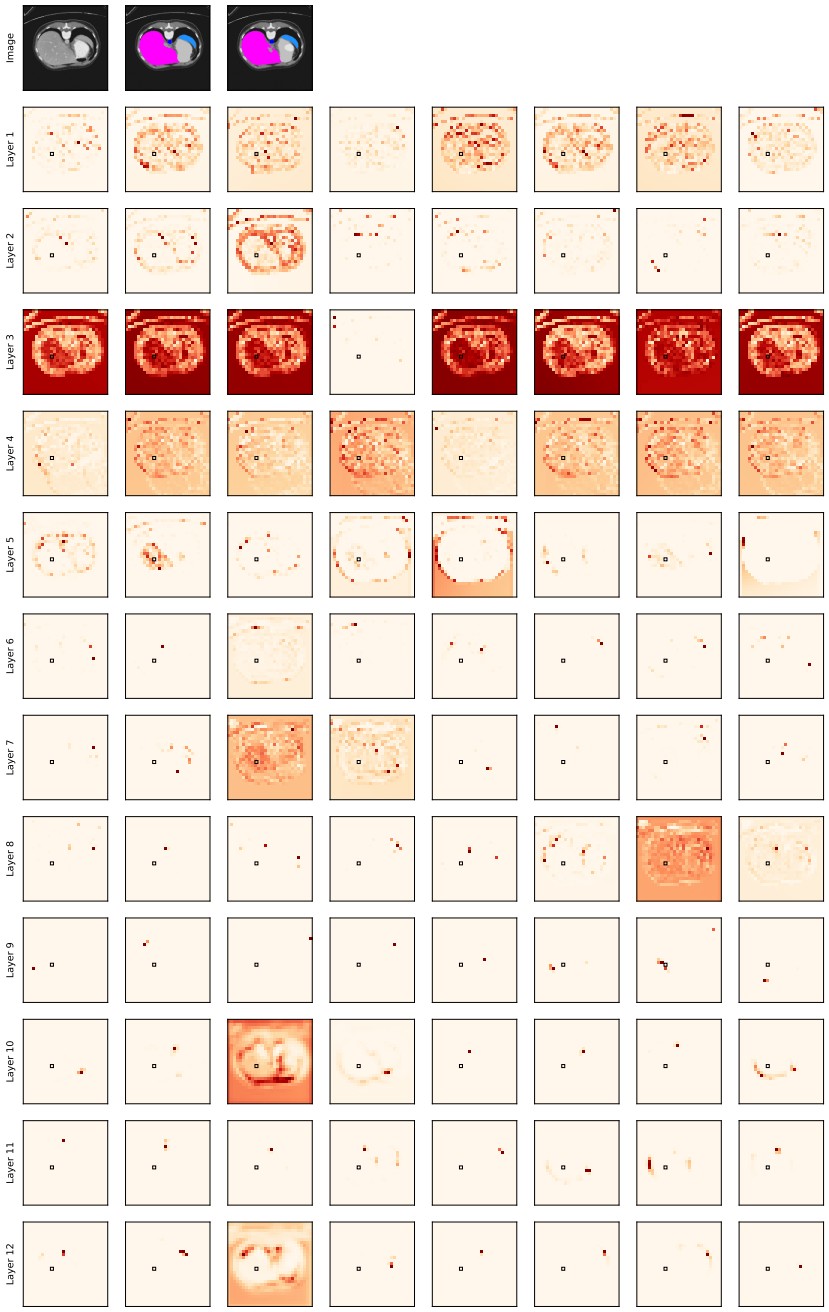

Figure 5: Attention probability of our 12 class-aware transformer layers, each with 8 heads. The black box marks the query patch. The input image, ground truth and predicted label are shown on the first row.

the semantically strong multi-scale feature pyramid from a single-scale input. The major differences between FPN and our work are as follows:

- The former utilizes a CNN-based decoder (FPN [16]), and ours uses an All-MLP-based decoder. In particular, our motivation comes from the observation that the attention of lower layers tends to be local, and those of the higher layers are highly non-local [17]. As the decoder design plays an important role in determining the semantic level of the latent representations [18] and Transformers have the larger receptive fields compared to CNNs, how to use large receptive fields to include context information is the key issue [17, 19–27].

Prior work [17] suggests that the use of MLP-based decoder design can be a very effective tool in learning additional contextual information to build powerful representations. The key idea is to essentially take benefits of the Transformer-induced features by leveraging the local attention at the lower layers and highly non-local (global) attention at the higher layers to formulate the powerful representations [17]. To this end, we utilize an MLP-based decoder instead of a CNN-based decoder to preserve more contextual information, specifically for medical imaging data, including more anatomical variances.

- We devise the class-aware transformer module to progressively learn interesting anatomical regions correlated with semantic structures of images, so as to guide the segmentation of objects or entities. We study the model's qualitative behavior through learnable sampling locations inside the class-aware module in Figure 4. As indicated, sampling locations are adaptively adjusted according to the interesting regions.

The table below shows the comparision results of using an `FPN` decoder, MLP-based decoder, and the class-aware transformer (CAT) module, all of which include the backbone feature extractor (`ResNet50`), on the Synapse multi-organ CT dataset. All the experiments are conducted under the same experimental setting in Section **??**. As we can see, adopting the MLP-based decoder can outperform the state-of-the-art `FPN` decoder in terms of DSC, Jaccard, 95HD, and ASD, respectively. Similarly, incorporating the CAT module can also consistently improve the segmentation performance by a large margin on the Synapse multi-organ CT dataset. The results prove the robustness of our MLP-based decoder and the effectiveness of our proposed CAT module for medical image segmentation.

Table 6: Ablation on Decoder Modules: `FPN` decoder [16]; MLP-based decoder; and Class-Aware Transformer (CAT) module.

| Encoder | Decoder | DSC | Jaccard | 95HD | ASD |
|---------|---------|-------|---------|-------|------|
| ResNet50 w/o CAT | FPN | 74.64 | 63.91 | 29.54 | 8.81 |
| ResNet50 w/ CAT | FPN | 78.11 | 65.63 | 28.06 | 8.08 |
| ResNet50 w/o CAT | MLP | 80.09 | 70.56 | 25.62 | 7.30 |
| ResNet50 w/ CAT | MLP | 82.17 | 73.22 | 16.20 | 4.28 |

## M    More Ablations on Segmentation Losses

To deal with the imbalanced medical image segmentation, Lin *et al.* [28] proposed Focal loss in terms of the standard cross entropy to address the extreme foreground-background class imbalance by focusing on the hard pixel examples. The table below shows the results of the loss function. We follow $\gamma = 2$ in the original paper. As we can see, the setting using Focal loss and the other (*i.e.*, Dice + Cross-Entropy) achieve similar performances.

Table 7: Ablation on Segmentation Losses: Focal loss [28]; Dice loss; and Cross-Entropy loss.

| Model | DSC | Jaccard | 95HD | ASD |
|-------|-------|---------|-------|------|
| Focal loss [28] | 82.08 | 73.52 | 16.14 | 4.99 |
| Dice + Focal loss [28] | 81.88 | 72.94 | 16.52 | 5.00 |
| Dice + Cross-Entropy loss (ours) | 82.17 | 73.22 | 16.20 | 4.28 |

## N    More Ablations on Sampling Modules

In this section, we investigate the effect of recent state-of-the-art sampling modules [29–31]. However, the motivation and the sampling strategy are different from these works [29–31]. Our motivation comes from the accurate and reliable clinical diagnosis that rely on the meaningful radiomic features from the correct "region of interest" instead of other irrelevant parts [32–35]. The process of extracting different radiomic features from medical images is done in a progressive and adaptive manner [33, 34].

`DCN` [29] proposed to learn 2D spatial offsets to enable the CNN-based model to generalize the capability of regular convolutions. Because CNNs only have limited receptive fields compared to Transformers, `DCN` focuses on local information around a certain point of interest. In contrast, our `CATformer`/`CASTformer` take benefits of the Transformer-induced features by leveraging the local

attention at the lower layers and highly non-local (global) attention at the higher layers to formulate the powerful representations.

`Deformable DETR` [30] incorporated the deformation attention to focus on a sparse set of keys (*i.e.*, global keys are not shared among visual tokens). This is particularly useful for its original experiment setup on object detection. Since there are only a handful of query features corresponding to potential object classes, deformable DETR learns different attention locations for each class. In contrast, our approach aims at refining the anatomical tokens for medical image segmentation. To this end, we proposed to iteratively and adaptively focus on the most discriminative region of interests. This essentially allows us to obtain effective anatomical features from spatial attended regions within the medical images, so as to guide the segmentation of objects or entities

`DAT` [31] introduced deformable attention to make use of global information (*i.e.*, global keys are shared among visual tokens) by placing a set of the supporting points uniformly on the feature maps. In contrast, our approach introduces an iterative and progressive sampling strategy to capture the most discriminative region and avoid over-partition anatomical features.

The table below shows the comparison results between `DCN` [29], `Deformable DETR` [30], `DAT` [31], and ours (`CATformer/CASTformer`) on the Synapse multi-organ CT dataset. As we can see, our approach (*i.e.*, `CATformer/CASTformer`) can outperform existing state-of-the-art models, *i.e.*,`DCN` [29], and `Deformable DETR`.

Table 8: Ablation on Sampling Module: `DCN` [29], `Deformable DETR` [30], `DAT` [31], and ours (`CATformer/CASTformer`).

| Model | DSC | Jaccard | 95HD | ASD |
|---|---|---|---|---|
| `DCN` [29] | 73.19 | 62.81 | 33.46 | 10.22 |
| `Deformable DETR` [30] | 79.13 | 66.58 | 30.21 | 8.65 |
| `DAT` [31] | 80.34 | 68.15 | 26.14 | 7.76 |
| `CATformer` (ours) | 82.17 | 73.22 | 16.20 | 4.28 |
| `CASTformer` (ours) | 82.55 | 74.69 | 22.73 | 5.81 |

## O  More Ablations on Architecture Backbone

In this section, we conduct the ablation study on the Synapse multi-organ CT dataset to compare our approach with the recent state-of-the-art architecture (`SwinUnet`) [11]. The table below shows the results of our proposed architecture (*e.g.*, Swin-class-aware transformer (Swin-CAT) module, multi-scale feature extraction module) are superior compared to the other state-of-the-art method on the Synapse multi-organ CT dataset. All the experiments are conducted under the same experimental setting in Section **??**. For brevity, we refer our `CATformer` and `CASTformer` using `SwinUnet` as the backbone to `Swin-CATformer` and `Swin-CASTformer`. As we can see, using `SwinUnet` as the backbone, the following observations can be drawn: (1) "w/ pre-trained" consistently achieves significant performance gains compared to the "w/o pre-trained", which demonstrates the effectiveness of the pre-training strategy; (2) we can find that incorporating the adversarial training can boost the segmentation performance, which suggests the effectiveness of the adversarial training strategy; and (3) our `Swin-CASTformer` with different modules can also achieves consistently improved performance. The results prove the superiority of our proposed method on the medical image segmentation task.

Table 9: Effect of transfer learning in our `Swin-CATformer` and `Swin-CASTformer` on the Synapse multi-organ dataset.

| Model | DSC | Jaccard | 95HD | ASD |
|---|---|---|---|---|
| • `Swin-CATformer` (w/o pre-trained) | 76.82 | 65.44 | 29.58 | 8.58 |
| • `Swin-CATformer` (w/ pre-trained) | 80.19 | 70.61 | 22.66 | 6.02 |
| ○ `Swin-CASTformer` (*both* w/o pre-trained) | 71.67 | 61.08 | 43.01 | 13.21 |
| ○ `Swin-CASTformer` (*only* w/ pre-trained $D$) | 76.55 | 64.27 | 34.62 | 12.13 |
| ○ `Swin-CASTformer` (*only* w/ pre-trained $G$) | 77.12 | 65.39 | 30.99 | 11.00 |
| ○ `Swin-CASTformer` (*both* w/ pre-trained) | 80.49 | 71.19 | 23.94 | 6.91 |

## References

[1] Ekin D Cubuk, Barret Zoph, Jonathon Shlens, and Quoc V Le. Randaugment: Practical automated data augmentation with a reduced search space. In *CVPR Workshops*, 2020.

Table 10: Ablation on model component: Baseline; `Swin-CATformer` w/o Swin-CAT; `Swin-CATformer` w/o multi-scale feature extraction; and `Swin-CASTformer`.

| Model | DSC | Jaccard | 95HD | ASD |
|---|---|---|---|---|
| Baseline | 76.33 | 65.64 | 27.16 | 8.32 |
| ● `Swin-CATformer` w/o Swin-CAT | 77.76 | 68.47 | 25.26 | 7.15 |
| ● `Swin-CATformer` w/o multi-scale feature extraction | 78.45 | 78.26 | 24.94 | 7.08 |
| ● `Swin-CATformer` | 80.19 | 70.61 | 22.66 | 6.02 |
| ○ `Swin-CASTformer` | 80.49 | 71.19 | 23.94 | 6.91 |

[2] Christian Szegedy, Vincent Vanhoucke, Sergey Ioffe, Jonathon Shlens, and Zbigniew Wojna. Rethinking the inception architecture for computer vision. In *IEEE Conference on Computer Vision and Pattern Recognition (CVPR)*, 2016.

[3] Hongyi Zhang, Moustapha Cisse, Yann N Dauphin, and David Lopez-Paz. mixup: Beyond empirical risk minimization. In *International Conference on Learning Representations (ICLR)*, 2018.

[4] Sangdoo Yun, Dongyoon Han, Seong Joon Oh, Sanghyuk Chun, Junsuk Choe, and Youngjoon Yoo. Cutmix: Regularization strategy to train strong classifiers with localizable features. In *IEEE International Conference on Computer Vision (ICCV)*, 2019.

[5] Boris T Polyak and Anatoli B Juditsky. Acceleration of stochastic approximation by averaging. *SIAM Journal on Control and Optimization*, 1992.

[6] Olaf Ronneberger, Philipp Fischer, and Thomas Brox. U-net: Convolutional networks for biomedical image segmentation. In *International Conference on Medical Image Computing and Computer-Assisted Intervention (MICCAI)*, 2015.

[7] Jo Schlemper, Ozan Oktay, Michiel Schaap, Mattias Heinrich, Bernhard Kainz, Ben Glocker, and Daniel Rueckert. Attention gated networks: Learning to leverage salient regions in medical images. *Medical Image Analysis*, 2019.

[8] Sixiao Zheng, Jiachen Lu, Hengshuang Zhao, Xiatian Zhu, Zekun Luo, Yabiao Wang, Yanwei Fu, Jianfeng Feng, Tao Xiang, Philip HS Torr, et al. Rethinking semantic segmentation from a sequence-to-sequence perspective with transformers. In *IEEE Conference on Computer Vision and Pattern Recognition (CVPR)*, 2021.

[9] Yutong Xie, Jianpeng Zhang, Chunhua Shen, and Yong Xia. Cotr: Efficiently bridging cnn and transformer for 3d medical image segmentation. In *International Conference on Medical Image Computing and Computer-Assisted Intervention (MICCAI)*, 2021.

[10] Jieneng Chen, Yongyi Lu, Qihang Yu, Xiangde Luo, Ehsan Adeli, Yan Wang, Le Lu, Alan L Yuille, and Yuyin Zhou. Transunet: Transformers make strong encoders for medical image segmentation. In *International Conference on Medical Image Computing and Computer-Assisted Intervention (MICCAI)*, 2021.

[11] Hu Cao, Yueyue Wang, Joy Chen, Dongsheng Jiang, Xiaopeng Zhang, Qi Tian, and Manning Wang. Swin-unet: Unet-like pure transformer for medical image segmentation. *arXiv preprint arXiv:2105.05537*, 2021.

[12] Ishaan Gulrajani, Faruk Ahmed, Martin Arjovsky, Vincent Dumoulin, and Aaron Courville. Improved training of wasserstein gans. In *Advances in Neural Information Processing Systems (NeurIPS)*, 2017.

[13] Ian Goodfellow, Jean Pouget-Abadie, Mehdi Mirza, Bing Xu, David Warde-Farley, Sherjil Ozair, Aaron Courville, and Yoshua Bengio. Generative adversarial nets. In *Advances in Neural Information Processing Systems (NeurIPS)*, 2014.

[14] Mario Lucic, Karol Kurach, Marcin Michalski, Sylvain Gelly, and Olivier Bousquet. Are gans created equal? a large-scale study. *arXiv preprint arXiv:1711.10337*, 2017.

[15] Xudong Mao, Qing Li, Haoran Xie, Raymond YK Lau, Zhen Wang, and Stephen Paul Smolley. Least squares generative adversarial networks. In *IEEE International Conference on Computer Vision (ICCV)*, 2017.

[16] Tsung-Yi Lin, Piotr Dollár, Ross Girshick, Kaiming He, Bharath Hariharan, and Serge Belongie. Feature pyramid networks for object detection. In *IEEE Conference on Computer Vision and Pattern Recognition (CVPR)*, 2017.

[17] Enze Xie, Wenhai Wang, Zhiding Yu, Anima Anandkumar, Jose M Alvarez, and Ping Luo. Segformer: Simple and efficient design for semantic segmentation with transformers. *arXiv preprint arXiv:2105.15203*, 2021.

[18] Kaiming He, Xinlei Chen, Saining Xie, Yanghao Li, Piotr Dollár, and Ross Girshick. Masked autoencoders are scalable vision learners. In *IEEE Conference on Computer Vision and Pattern Recognition (CVPR)*, 2022.

[19] Fisher Yu and Vladlen Koltun. Multi-scale context aggregation by dilated convolutions. *arXiv preprint arXiv:1511.07122*, 2015.

[20] Chao Peng, Xiangyu Zhang, Gang Yu, Guiming Luo, and Jian Sun. Large kernel matters–improve semantic segmentation by global convolutional network. In *IEEE Conference on Computer Vision and Pattern Recognition (CVPR)*, 2017.

[21] Liang-Chieh Chen, Yukun Zhu, George Papandreou, Florian Schroff, and Hartwig Adam. Encoder-decoder with atrous separable convolution for semantic image segmentation. In *European Conference on Computer Vision (ECCV)*, 2018.

[22] Chenyu You, Junlin Yang, Julius Chapiro, and James S. Duncan. Unsupervised wasserstein distance guided domain adaptation for 3d multi-domain liver segmentation. In *Interpretable and Annotation-Efficient Learning for Medical Image Computing*, pages 155–163. Springer International Publishing, 2020.

[23] Chenyu You, Ruihan Zhao, Lawrence Staib, and James S Duncan. Momentum contrastive voxel-wise representation learning for semi-supervised volumetric medical image segmentation. *arXiv preprint arXiv:2105.07059*, 2021.

[24] Chenyu You, Yuan Zhou, Ruihan Zhao, Lawrence Staib, and James S Duncan. Simcvd: Simple contrastive voxel-wise representation distillation for semi-supervised medical image segmentation. *IEEE Transactions on Medical Imaging*, 2022.

[25] Chenyu You, Weicheng Dai, Lawrence Staib, and James S Duncan. Bootstrapping semi-supervised medical image segmentation with anatomical-aware contrastive distillation. *arXiv preprint arXiv:2206.02307*, 2022.

[26] Chenyu You, Jinlin Xiang, Kun Su, Xiaoran Zhang, Siyuan Dong, John Onofrey, Lawrence Staib, and James S Duncan. Incremental learning meets transfer learning: Application to multi-site prostate mri segmentation. *arXiv preprint arXiv:2206.01369*, 2022.

[27] Chenyu You, Weicheng Dai, Fenglin Liu, Haoran Su, Xiaoran Zhang, Lawrence Staib, and James S Duncan. Mine your own anatomy: Revisiting medical image segmentation with extremely limited labels. *arXiv preprint arXiv:2209.13476*, 2022.

[28] Tsung-Yi Lin, Priya Goyal, Ross Girshick, Kaiming He, and Piotr Dollár. Focal loss for dense object detection. In *IEEE International Conference on Computer Vision (ICCV)*, 2017.

[29] Jifeng Dai, Haozhi Qi, Yuwen Xiong, Yi Li, Guodong Zhang, Han Hu, and Yichen Wei. Deformable convolutional networks. In *IEEE International Conference on Computer Vision (ICCV)*, 2017.

[30] Xizhou Zhu, Weijie Su, Lewei Lu, Bin Li, Xiaogang Wang, and Jifeng Dai. Deformable detr: Deformable transformers for end-to-end object detection. *arXiv preprint arXiv:2010.04159*, 2020.

[31] Zhuofan Xia, Xuran Pan, Shiji Song, Li Erran Li, and Gao Huang. Vision transformer with deformable attention. In *IEEE Conference on Computer Vision and Pattern Recognition (CVPR)*, pages 4794–4803, 2022.

[32] Alex Zwanenburg, Martin Vallières, Mahmoud A Abdalah, Hugo JWL Aerts, Vincent Andrearczyk, Aditya Apte, Saeed Ashrafinia, Spyridon Bakas, Roelof J Beukinga, Ronald Boellaard, et al. The image biomarker standardization initiative: standardized quantitative radiomics for high-throughput image-based phenotyping. *Radiology*, 295(2):328–338, 2020.

[33] Joost JM Van Griethuysen, Andriy Fedorov, Chintan Parmar, Ahmed Hosny, Nicole Aucoin, Vivek Narayan, Regina GH Beets-Tan, Jean-Christophe Fillion-Robin, Steve Pieper, and Hugo JWL Aerts. Computational radiomics system to decode the radiographic phenotype. *Cancer research*, 77(21):e104–e107, 2017.

[34] Ahmet Saygılı. A new approach for computer-aided detection of coronavirus (covid-19) from ct and x-ray images using machine learning methods. *Applied Soft Computing*, 105:107323, 2021.

[35] Feng Shi, Liming Xia, Fei Shan, Bin Song, Dijia Wu, Ying Wei, Huan Yuan, Huiting Jiang, Yichu He, Yaozong Gao, et al. Large-scale screening to distinguish between covid-19 and community-acquired pneumonia using infection size-aware classification. *Physics in medicine & Biology*, 66(6):065031, 2021.