# OpenReview forum: " Class-Aware Adversarial Transformers for Medical Image Segmentation "
_NeurIPS.cc/2022/Conference — NeurIPS 2022 Accept_

### Official Review · Reviewer_oVPV · 2022-07-11

**Rating:** 6
**Confidence:** 5
**Soundness:** 3 good
**Presentation:** 2 fair
**Contribution:** 3 good

**Summary:**

This paper proposed Transformer based segmentation model for medical image called CASTformer. Compared to the other segmentation algorithms, it utilizes 1) the multi-scale approach with CNN+Transformer hybrid architecture, 2) the class-aware transformer module for progressive sampling strategy, and the adversarial training scheme. Also, the pre-trained model in computer vision dataset helps to start their training in good starting point. The results show that CASTformer improves the segmentation performance impressively.

**Questions:**

- Please refer and explain the differences between your model and the deformable sampling CNN/Transformers: DCN(Deformable convolutional networks), DAT(Vision Transformer with Deformable Attention), Deformable DETR(Deformable DETR: Deformable transformers for end-to-end object detection), and etc. They are using same concept for feature sampling with irregular grid. The comparison results with the other deformable sampling module and the reason why it works better might be helpful for the readers.

- Is there any performance drop using the bilinear interpolation in the feature domain? How much the performance drop for discrete sampling such as nearest neighbor sampling? I have some worries about the existence of continuity in 2D grid features. Please, refer some theoretical background proof or practical evidence for that.

- Fair comparison between pre-trained model and from-scratch model.
I wonder the epoch numbers for the training of from-pretrained model and from-scratch model is same or not.

- Ablation study.
As you shown in the analysis, the proposed train processes (such as pre-training on natural image dataset and adversarial loss) seems very effective to improve the performance dramatically. Then, to support the proposed architectural excellence, it is necessary to compare between SwinUnet(or other SOTA architecture) with the proposed  training process(adversarial loss+pre-training with computer vision dataset) and the proposed CASTformer. It is necessary to show that the proposed architecture (class-aware Transformer module, Multi-scale feature extraction, etc) is superior to others.


**Limitations:**


- Why the term ‘generative’ is used for your model? Does your model generate the image which not exists in the input image? The proposed method utilizes discriminator to improve the performance of segmentation with adversarial auxiliary task for the ROI reconstruction not the generation of some parts not in the input. The outputs are the reconstruction of the attended region of input image, not the generation. In my opinion, the term ‘generative’ should be used at a minimum to maximize the persuasive power of your claim because some (quiet large portion of the medical field) peoples in real clinic have a lot of worries to use generative model for the medical imaging.


- The term ‘class-aware’ is confusing.
Some readers in computer vision field or machine learning field might expect that the proposed ‘class-aware’ module dynamically/adaptively utilizes the class information like in StyleGAN. However, the authors used the term ‘class-aware’ for the module which samples discriminative locations by iteratively estimated the sampling offsets based on the current input features.
The author might say the class-aware Transformer module samples different point according to the type of organ. However, does the class-aware module have ability to distinguish kidney/liver/spleen? I think the sampling module does not distinguish the type of organ. It focuses on the current feature pixel is related/connected to the other feature pixel or not according to the global information.
To support your claim, the author should provide the experimental support which adding a linear layer on the proposed class-aware module with the freezed weight has some ability to classify the organ type class.


**Strengths And Weaknesses:**

- (significance) CASTformer shows the dramatic improvement of segmentation performance with large gap to the other algorithms. The author proposed better training processes to utilized the adversarial loss and the computer vision dataset as pre-training to improve the performance of segmentation task for the (relatively-) limited medical dataset.

- (originality & quality) Some reader might claim that they combined many concepts already exists in computer vision field such as multi-scale feature extraction, feature sampling, adversarial loss and transfer-learning. However, it is important that they make that work and it reaches the dramatic performance improvement on their task.

- (clarity)  They support various ablation studies and comparison with many deep network algorithms. However, there are some lack of clarity and I asked those questions in the followings.

---

> ### Author Response · Authors · 2022-08-02
> **Response to Reviewer oVPV (Part 1)**
>
> We thank the reviewer for acknowledging our contribution to the medical image analysis field, appreciating the dramatic performance improvement on our multi-class medical segmentation task, and providing constructive suggestions for the presentation of our work! We’ve made a substantial revision to the paper, which addresses all the issues, with emphasis on clarity of the explanation of our work. If you have further concerns, please feel free to contact us.
>
> > **Q1**: Comparision between your model and recent SOTA methods.
>
> **A1**: Thank you for the great suggestion! We follow your constructive advice to evaluate these works. Our work is related to DCN(Deformable convolutional networks) [88], Deformable DETR(Deformable DETR: Deformable transformers for end-to-end object detection) [89], DAT(Vision Transformer with Deformable Attention) [90]. However, our goals and motivations are different. In particular, the motivation and the sampling strategy are different from these works [88-90]. Our motivation comes from the accurate and reliable clinical diagnosis that rely on the meaningful radiomic features from the correct “region of interest” instead of other irrelevant parts [91-94]. The process of extracting different radiomic features from medical images is in the progressive and adaptive manner [92-93]. We have highlighted the comparison in Appendix Section-N in the latest manuscript.
>
> DCN [88] proposed to learn 2D spatial offsets to enable the CNN-based model to generalize the capability of regular convolutions. Because CNNs only have limited receptive fields compared to Transformers, DCN focuses on local information around a certain point of interest. In contrast, our CATformer/CASTformer take benefits of the Transformer-induced features by leveraging the local attention at the lower layers and highly non-local (global) attention at the higher layers to formulate the powerful representations.
>
> Deformable DETR [89] incorporated the deformation attention to focus on a sparse set of keys (i.e., global keys NOT shared among visual tokens). This is particularly useful for its original experiment setup on object detection. Since there are only a handful of query features corresponding to potential object classes, deformable DETR learns different attention locations for each class. In contrast, our approach aims at refining the anatomical tokens for medical image segmentation. To this end, we proposed to iteratively and adaptively focus on the most discriminative region of interests. This essentially allows us to obtain effective anatomical features from spatial attended regions within the medical images, so as to guide the segmentation of objects or entities
>
> DAT [90] introduces deformable attention to make use of global information (i.e., global keys shared among visual tokens) by placing a set of the supporting points uniformly on the feature maps. In contrast, our approach introduces an iterative and progressive sampling strategy to capture the most discriminative region and avoid over-partition anatomical features.
>
> Following your constructive advice, the table below shows the comparison results between DCN, Deformable DETR, DAT, and ours (CATformer/CASTformer) on the Synapse multi-organ CT dataset.
>
> | Model  | DSC  | Jaccard  | 95HD  | ASD  |
> | :-----------  | :-----------:  | :-----------:  | :-----------:  | :-----------:  |
> |  DCN  |  73.19  |  62.81  |  33.46  |  10.22  |
> |  Deformable DETR  |  79.13  |  66.58  |  30.21  |  8.65  |
> |  DAT  |  80.34  |  68.15  |  26.14  |  7.76  |
> |  CATformer (ours)  |  82.17  |  73.22  |  **16.20**  |  **4.28**  |
> |  CASTformer (ours)  |  **82.55**  |  **74.69**  |  22.73  |  5.81  |
>
> As we can see, our approach (i.e., CATformer/CASTformer) can outperform existing state-of-the-art models, i.e., DCN, Deformable DETR, and DAT.

---

> > ### Author Response · Authors · 2022-08-02
> > **Response to Reviewer oVPV (Part 2)**
> >
> > Thanks for your helpful comments! If you have further concerns, please feel free to contact us.
> >
> > > **Q2**: Is there any performance drop using the bilinear interpolation in the feature domain? How much the performance drop for discrete sampling such as nearest neighbor sampling? I have some worries about the existence of continuity in 2D grid features. Please, refer some theoretical background proof or practical evidence for that.
> >
> > **A2**: Thank you for the great suggestion! We agree with your point that using the bilinear interpolation in the feature space might cause performance drops [8]. But the key motivation for using bilinear interpolation in this study is to build a differentiable sampling mechanism, which enables the backpropagation of the loss not only to the feature maps but also to the sampling grid coordinates [8]. A similar practice can also be applied in recent works [33, 67, 88-90]. On the other hand, the major difference between nearest neighbor sampling and the bilinear interpolation is as follows: nearest neighbor sampling first *quantizes* a floating-number sampling location $\textbf{s}_t$ to the discrete granularity of the input feature map $\textbf{F}_t$. Such quantizations introduce misalignments between sampling locations and the extracted feature maps, leading to a large negative effect on the pixel-wise prediction tasks [67]. In contrast, the bilinear interpolation allows us to remove the harsh quantization by computing the exact floating-number (fractional) values of the sampled location in order to avoid discontinuities in the sampling function.
> >
> > We compare our bilinear interpolation with the nearest neighbor on the Synapse multi-organ CT dataset. All the experiments are conducted under the same experimental setting in Section 4.
> >
> > | Model  | DSC  | Jaccard  | 95HD  | ASD  |
> > | :-----------  | :-----------:  | :-----------:  | :-----------:  | :-----------:  |
> > |  CATformer (w/o nearest-neighbor)  |  79.81  |  69.64  |  28.97  |  8.31  |
> > |  CATformer (w/ bilinear interpolation)  |  **82.17**  |  **73.22**  |  **16.20**  |  **4.28**  |
> >
> > As we can see, adopting the bilinear interpolation can improve the segmentation accuracy compared to the nearest neighbor by a large margin. This is because the bilinear interpolation is more sensitive to localization accuracy. This also highlights that proper interpolation is critical to segmentation performance.
> >
> > > **Q3**: Fair comparison between pre-trained model and from-scratch model.
> >
> > **A3**: We train the model for 100 epochs in the pre-trained stage and train the model for 300 epochs. As for the from-scratch model, we train the model for 400 epochs. Therefore, it's a fair comparison between the from-pretrained model, and the from-scratch model on the Synapse multi-organ CT dataset. All the experiments are conducted under the same experimental setting in Section 4. The following Table shows that using the pre-training strategy is able to provide us with a good set of initial parameters that quickly adapt to new downstream medical segmentation tasks without re-building billions of anatomical representations, and further boost the performance.
> >
> > | Model  | DSC  | Jaccard  | 95HD  | ASD  |
> > | :-----------  | :-----------:  | :-----------:  | :-----------:  | :-----------:  |
> > |  CATformer (w/o pre-trained)  |  74.84  |  65.61  |  31.81  |  7.23  |
> > |  CATformer (w/ pre-trained)  |  **82.17**  |  **73.22**  |  **16.20**  |  **4.28**  |
> >
> > > **Q4**: Why the term ‘generative’ is used for your model? Does your model generate the image which not exists in the input image?
> >
> > **A4**: Thank you! We appreciate the reviewer for acknowledging the advantage of our proposed method for the improved segmentation performance. Indeed the images used for adversarial training are pixels from the input image with organs, as opposed to being generated from some latent distribution. We agree with the reviewer and appreciate them for raising the concern about the wording. We have rephrased it in our latest revision.

---

> > > ### Author Response · Authors · 2022-08-02
> > > **Response to Reviewer oVPV (Part 3)**
> > >
> > > Thanks for your helpful comments! If you have further concerns, please feel free to contact us.
> > >
> > > > **Q5**: Ablation study. As you shown in the analysis, the proposed train processes (such as pre-training on natural image dataset and adversarial loss) seems very effective to improve the performance dramatically. Then, to support the proposed architectural excellence, it is necessary to compare between SwinUnet(or other SOTA architecture) with the proposed training process(adversarial loss+pre-training with computer vision dataset) and the proposed CASTformer. It is necessary to show that the proposed architecture (class-aware Transformer module, Multi-scale feature extraction, etc) is superior to others.
> > >
> > > **A5**: Thank you for pointing out the comparison with the other SOTA architectures (e.g., SwinUnet) with the proposed training process and the proposed CASTformer. Following your great advice, we conducted the ablation study on the Synapse multi-organ CT dataset. The table below shows the results of our proposed architecture (e.g., class-aware transformer module, multi-scale feature extraction) are superior compared to the other state-of-the-art method on the Synapse multi-organ CT dataset. All the experiments are conducted under the same experimental setting in Section 4. For brevity, we refer our CATformer and CASTformer using SwinUnet as the backbone to Swin-CATformer and Swin-CASTformer. We have highlighted the comparison in Appendix Section-O in the latest manuscript.
> > >
> > > | Method  | DSC  | Jaccard  | 95HD  | ASD  |
> > > | :-----------  | :-----------:  | :-----------:  | :-----------:  | :-----------:  |
> > > |  Swin-CATformer (w/o pre-trained)  |  76.82  | 65.44  |  29.58  |  8.58  |
> > > |  Swin-CATformer (w/ pre-trained)  | 80.19  |  70.61  |  22.66  |  6.02  |
> > > |    |    |    |    |    |    |
> > > |  Swin-CASTformer (both w/o pre-trained)  | 71.67  |  61.08  |  43.01  |  13.21  |
> > > |  Swin-CASTformer (*only* w/ pre-trained $D$)  |  76.55  |  64.27  |  34.62  |  12.13  |
> > > |  Swin-CASTformer (*only* w/ pre-trained $G$)  |  77.12  |  65.39  |  30.99  |  11.00  |
> > > |  Swin-CASTformer (*both* w/ pre-trained)  |  80.49  |  71.19  |  23.94  |  6.91  |
> > >
> > > | Model  | DSC  | Jaccard  | 95HD  | ASD  |
> > > | :-----------  | :-----------:  | :-----------:  | :-----------:  | :-----------:  |
> > > |  Baseline (SwinUnet)  |  76.33  |  65.64  |  27.16  |  8.32  |
> > > |  Swin-CATformer (w/o Swin-class-aware transformer module)  |  77.76  |  68.47  |  25.26  |  7.15  |
> > > |  Swin-CATformer (w/o multi-scale feature extraction)  |  78.45  |  78.26  |  24.94  |  7.08  |
> > > |  Swin-CATformer  | 80.19  |  70.61  |  22.66  |  6.02  |
> > > |  Swin-CASTformer  |  80.49  |  71.19  |  23.94  |  6.91  |
> > >
> > > As we can see, using Swin-Unet as the baskbone, the following observations can be drawn: (1) “w/ pre-trained” consistently achieves significant performance gains compared to the  “w/o pre-trained”, which demonstrates the effectiveness of the pre-training strategy; (2) we can find that incorporating the adversarial training can boost the segmentation performance, which suggests the effectiveness of the adversarial training strategy; and (3) our Swin-CASTformer with different modules can also achieves consistently improved performance. The results prove the superiority of our proposed method on the medical image segmentation task.
> > >
> > >
> > > > **Q6**: The term ‘class-aware’ is confusing.
> > >
> > > **A6**: Thank you for the great suggestion. The name `class-aware' comes from the observation that the progressive sampling component results in pixel samples that concentrate at nearby organ locations, even when they belong to different classes. This is best illustrated in Figure 8 (Line 662 - 663), where the right kidney (cyan) and liver (magenta) are adjacent: we can see the model learns to attend to both regions. Since the model tends to move to discriminative organ regions instead of the background, we refer to our model as class-aware or organ-aware. In addition, when two organs are adjacent, we observe that the samples near the boundary often move to their corresponding organ instead of crossing the boundary. In this study, we don’t expect the sampling module to yield information for classification, as it might only need to move samples to proper attention areas. However, we totally agree with the reviewer that additional quantitative proof would solidify the term. For example, we could analyze the offsets' behavior along organ boundaries. Due to the limited response period and computational resources, we will work on it in future works.
> > >
> > > Overall, we’ve made a substantial revision to the paper, which addresses all the issues, with emphasis on the clarity of the explanation of our method, and proper choice of baselines for comparison. We hope that the changes will make this work in better shape for publication. Please feel free to contact us for further concerns.

---

> ### Author Response · Authors · 2022-08-08
> **Look forward to further feedback**
>
> Dear Reviewer oVPV:
>
> Thanks again for all of your constructive suggestions, which have helped us improve the quality and clarity of the paper!
>
> Since the author-reviewer discussion period will end soon in 2 days, we appreciate it if you take the time to read our rebuttal and give us some feedback. Please don't hesitate to let us know if there are any additional clarifications or experiments that we can offer, as we would love to convince you of the merits of the paper. We appreciate your suggestions. If our response resolves your concerns, we kindly ask you to consider raising the rating of our work.
>
> Thanks for your time and efforts!
>
> Best,
>
> Authors of Paper4486

---

> > ### Comment · Reviewer_oVPV · 2022-08-10
> > **Response to the author**
> >
> > Dear authors,
> >
> > I thank the authors for responding to the concerns and questions I have raised.
> > I carefully read the rebuttals and some of them convince me.
> > I would like to increase my score to weak acceptance based on my evaluation.
> >
> > Thank you for your replies and I have no more concerns.

---

### Official Review · Reviewer_YJsa · 2022-07-11

**Rating:** 7
**Confidence:** 3
**Soundness:** 3 good
**Presentation:** 3 good
**Contribution:** 3 good

**Summary:**

The authors introduce an effective generative adversarial transformer called CASTformer for 2D medical image segmentation. The main idea behind the design principle is to integrate the multi-scale pyramid structure to capture rich global spatial information as well as local multi-scale context information. The suggested class-aware transformer module enables CASTformer to discover useful aspects of objects incrementally and selectively. The generator-discriminator design is used to improve segmentation performance, enabling the transformer-based discriminator to capture both low-level anatomical information and high-level semantics.

**Questions:**

- Could you please check the reference papers and make comments on it with respect to your work?

**Limitations:**

Due to nature of the work, as per authors, research will not pose significant risks of societal harm to society. Rather, the study has the potential to positively contribute to a number of real-world clinical applications possibly.

**Strengths And Weaknesses:**

The proposed generative adversarial transformer called CASTformer for 2D medical image segmentation sounds interesting and useful. Both quantitative and qualitative results show the effectiveness of the proposed model. A number of ablations are performed to show that the suggested mechanisms are worth considering.

The authors provide clear justification for their work. This is a well-written manuscript and easy to follow indeed. The supplementary improves the overall attractiveness of the manuscript as well. The suggested mechanism is straightforward but effective. The authors validate their claims through extensive experiments and significant performance gain.

The study leverages pyramid structure to construct multi-scale representations and handle multi-scale variations. Some of the ideas here may be familiar to readers of the following papers [1, 2]. If these studies have any relevance to the topic at hand, it would be great if the authors would highlight it.

[1] T.-Y. Lin, P. Dollar, R. Girshick, K. He, B. Hariharan, and S. Belongie. Feature pyramid networks for object detection. In CVPR, 2017.

[2] T. -Y. Lin, P. Goyal, R. Girshick, K. He and P. Dollár, "Focal Loss for Dense Object Detection," in IEEE Transactions on Pattern Analysis and Machine Intelligence, vol. 42, no. 2, pp. 318-327, 1 Feb. 2020, doi: 10.1109/TPAMI.2018.2858826.

---

> ### Author Response · Authors · 2022-08-02
> **Response to Reviewer YJsa**
>
> We thank the reviewer for acknowledging our contribution and providing suggestions for the presentation of our work! In particular, we agree that comparisons with the reference papers are necessary to position this work properly. If you have further concerns, please feel free to contact us.
>
> > **Q1**: Relevant papers.
>
> **A1**: Thank you for the great suggestion. We agree with your point that the relevant papers share some similarities with our work. We follow your constructive advice to evaluate these works. We have highlighted in Section “Method”, and have extended two sections (See Appendix - L and M) in the latest manuscript to analyze and discuss these works.
>
> We explore another state-of-the-art backbone proposed by Lin et al. [73], termed Feature Pyramid Network (FPN). FPN utilizes a top-down pyramid with lateral connections to construct the semantically strong multi-scale feature pyramid from a single-scale input. The major differences between FPN and our work are as follows:
>
> (1) The former utilizes a CNN-based decoder, and ours uses an All-MLP-based decoder. In particular, our motivation comes from the observation that the attention of lower layers tends to be local, and those of the higher layers are highly non-local [74]. As the decoder design plays an important role in determining the semantic level of the latent representations [9] and Transformers have the larger receptive fields compared to CNNs, how to use large receptive fields to include context information is the key issue [68-70,74]. The key idea is to essentially take benefits of the Transformer-induced features by leveraging the local attention at the lower layers and highly non-local (global) attention at the higher layers to formulate the powerful representations [74]. To this end, we utilize an MLP-based decoder for preserving more contextual information, specifically for medical imaging data, including more anatomical variances.
>
> (2) We devise the class-aware transformer module to progressively learn interesting anatomical regions correlated with semantic structures of images, so as to guide the segmentation of objects or entities. We study the model’s qualitative behavior through learnable sampling locations inside the class-aware module in Figure 8 (Line 705-706). As indicated, sampling locations are adaptively adjusted according to the interesting regions.
>
> The table below shows the results of using an FPN decoder, MLP-based decoder, and the class-aware transformer (CAT) module, both of which include the backbone feature extractor (ResNet50), on the Synapse multi-organ CT dataset. All the experiments are conducted under the same experimental setting in Section 4.
>
> | Encoder | Decoder  | DSC  | Jaccard  | 95HD  | ASD  |
> | :-----------  | :-----------:  | :-----------:  | :-----------:  | :-----------:  | :-----------:  |
> |  ResNet50 w/o CAT  |  FPN  |  74.64  | 63.91  |  29.54  |  8.81  |
> |  ResNet50 w/   CAT  |  FPN  | 78.11  |  65.63  |  28.06  |  8.08  |
> |  ResNet50 w/o CAT  |  MLP  | 80.09  |  70.56  |  25.62  |  7.30  |
> |  ResNet50 w/   CAT  |  MLP  |  82.17  |  73.22  |  16.20  |  4.28  |
>
> As we can see, adopting the MLP-based decoder can outperform the state-of-the-art FPN decoder in terms of DSC, Jaccard, 95HD, and ASD, respectively. Similarly, incorporating the CAT module can also consistently improve the segmentation performance by a large margin on the Synapse multi-organ CT dataset. The results prove the robustness of our MLP-based decoder and the effectiveness of our proposed CAT module for medical image segmentation.
>
> To deal with the imbalanced medical image segmentation, Lin et al. [72] proposed Focal loss in terms of the standard cross entropy to address the extreme foreground-background class imbalance by focusing on the hard pixel examples. The table below shows the results of the loss function. We follow $\gamma = 2$ in the original paper.
>
> |  Model  |  DSC  |  Jaccard  |  95HD  |  ASD  |
> |  :-----------  |  :-----------:  |  :-----------:  |  :-----------:  |  :-----------:  |
> |  Focal Loss  |  82.08  |  73.52  |  16.14  |  4.99  |
> |  Dice + Focal Loss  |  81.88  |  72.94  |  16.52  |  5.00  |
> |  Dice + Cross-Entropy (ours)  |  82.17  |  73.22  |  16.20  |  4.28  |
>
> As we can see, using Focal loss and Dice + Cross-Entropy achieve similar performances. Due to the limited response period and computational resources, we will investigate different hyperparameter settings in Focal loss. We will make corresponding revisions to cite and provide detailed comparisons of them.
>
> Overall, thank you again for your suggestions and review! We believe that the papers suggested by the reviewers contain solid contributions and are highly relevant to our work in some respects. We have included the references and discussion of these relevant papers in our latest revision. We hope that the revision puts our work in better shape for publication. Please feel free to contact us for further concerns.

---

> > ### Comment · Reviewer_YJsa · 2022-08-09
> > **Response to the authors' rebuttal**
> >
> > I appreciate the authors' responses as well as their additional experiments. My concern has been addressed and my overall opinion about the paper remains unchanged. Although the reviewer aCmr expressed concerns about the novelty and contribution of the submission, I believe the study has its own merits and incremental contributions and has the potential to benefit the relevant community.

---

> > > ### Author Response · Authors · 2022-08-09
> > > **Thank you again for your review and very valuable feedback!**
> > >
> > > We thank the reviewer for acknowledging the positive changes we have made to the paper. We are genuinely happy that our major revision with more explanations and experiments properly addresses fellow reviewers' feedbacks and has made a difference. We thank the reviewer again for the constructive feedback which helps shape this revision!

---

### Official Review · Reviewer_aCmr · 2022-07-12

**Rating:** 4
**Confidence:** 5
**Soundness:** 3 good
**Presentation:** 2 fair
**Contribution:** 2 fair

**Summary:**

This paper proposed a transformer-based model for medical image segmentation, which adopts the pyramid structure and adversarial training. The proposed method is validated on three benchmark datasets, and obtains advantaged results.

**Questions:**

As indicated in this paper, it is a pre-trained model on public computer vision datasets. In the experiments, which computer vision dataset is employed? For medical image segmentation task, is the pre-trained model necessary?

**Limitations:**

see weakness

**Strengths And Weaknesses:**

Strengths:
1. the class-aware transformer module is interesting.
2. the paper is well written, and there are careful experimental analysis conducted.

Weakness:
1. the novelty and contribution are limited. The key contribution is the class-aware transformer module, a revised transformer, to learn the class aware context. Pyramid structure and adversarial training are common approaches used in medical image analysis.
2.The motivation is unclear. Why the adversarial network is needed in this model?
3. the comparison of experimental results is unfair. The proposed model is equipped with newly-added CAT and GAN, which is a bigger model than others. Even the pre-trained model is compared with other models.

---

> ### Author Response · Authors · 2022-08-02
> **Response to Reviewer aCmr (Part 1)**
>
> Thanks for your helpful comments! If you have further concerns, please feel free to contact us.
>
> > **Q1**: Which computer vision dataset is employed?
>
> **A1**: In our experiments, we adopt the parameters pre-trained on ImageNet-21k to initialize our model.
>
> > **Q2**: Is the pre-trained model necessary?
>
> **A2**: In the medical imaging domain, the data are scattered in various hospitals. Thus, it’s rather difficult to construct a large dataset. On the other hand, it is noticed that the transformer-based methods generally require to be pre-trained on the large-scale datasets (e.g., ImageNet) to perform well [4,5]. To this end, it is necessary to use the pre-trained weights as a good starting point. Moreover, we further experimentally demonstrate the necessity of using the pre-trained model compared to the from-scratch model (please see Table 2 Line 282 - 283). We can see that using pre-trained weights to initialize the $G$ and $D$ can both contribute to the performance gains, which further justify our parameter initialization scheme. For your convenience, the following Table shows the comparison results of the pre-trained model and the from-scratch model on the Synapse multi-organ CT dataset.
>
> | Model  | DSC  | Jaccard  | 95HD  | ASD  |
> | :-----------  | :-----------:  | :-----------:  | :-----------:  | :-----------:  |
> |  CATformer (w/o pre-trained)  |  74.84  |  65.61  |  31.81  |  7.23  |
> |  CATformer (w/ pre-trained)  |  **82.17**  |  **73.22**  |  **16.20**  |  **4.28**  |
> |  |  |  |  |  |
> |  CASTformer (*both* w/o pre-trained in $G$ and $D$)  |  73.64  |  62.68  |  42.77  |  11.76  |
> |  CASTformer (*only* w/ pre-trained in $D$)  |  78.87  |  69.36  |  30.54  |  9.17  |
> |  CASTformer (*only* w/ pre-trained in $G$)  |  81.46  |  71.80  |  27.36  |  6.91  |
> |  CASTformer (*both* w/ pre-trained in $G$ and $D$)  |  **82.55**  |  **74.69**  |  **22.73**  |  **5.81**  |
>
> Given the above ablation study, we observe that using “w/ pre-trained” leads to higher accuracy than “w/o pre-trained” with significant improvements for the smaller sizes of datasets, suggesting that using “w/ pre-trained” provides us a good set of initial parameters for the medical image segmentation tasks. With using pre-trained weights, our CATformer outperforms the setting without using pre-trained weights by a large margin and achieves $7.33\%$ and $7.61\%$ absolute improvements in terms of Dice and Jaccard. Our CASTformer (“w/ pre-trained”) also yields big improvements ($+8.91\%$ and $+12.01\%$) in Dice and Jaccard. This suggests that the pre-trained model can contribute to satisfactory segmentation performance.
>
> > **Q3**: The comparison of experimental results is unfair.
>
> **A3**: It is worth noting that, in this study, all the other transformer-based methods [7,10,13,56] use the pre-trained weights of the ImageNet-21k dataset. Such practice is commonly-used in recent medical image segmentation models, which all utilize ImageNet-21k pre-trained parameters as a starting point. Therefore, it's a fair comparison with previous works in Tables 1 (Line 236 - 237), 6 (Line 667 - 668), and 7 (Line 671 - 672).
>
> > **Q4**: Limited novelty and contribution.
>
> **A4**: We focus on improving the interpretability of medical segmentation models, which is the key aspect of successful medical image analysis. We make the **first attempt** to build an adversarial training framework using a transformer-based architecture for solving this task, resulting in the **c**lass-**a**ware adver**s**arial **t**rans**former**s (CASTformer), which includes a class-aware transformer module to progressively learn interesting regions correlated with semantic structures of images. The key challenge of learning-based medical image segmentation is to address the issue in interpretability. To this end, we study the model’s qualitative behavior through learnable sampling locations inside the class-aware module in Figure 8 (Line 705 - 706). As indicated, sampling locations are adaptively adjusted according to the interesting regions.
>
> Therefore, this work could provide a good basis or starting point for the research of interpretable medical image segmentation. The good robustness of our proposed model to the pre-trained model and a relatively small medical dataset illustrates the benefits of leveraging pre-trained models from the computer vision domain and provides suggestions for future research that could be less susceptible to the confounding effects of training data from the natural image domain, enables our method to have the potential to be applied to other medical image analysis tasks such as medical image enhancement (CT/MRI/PET reconstruction) and registration, where the size of labeled medical image data could be much limited or even unavailable.

---

> > ### Author Response · Authors · 2022-08-02
> > **Response to Reviewer aCmr (Part 2)**
> >
> > Thanks for your helpful comments! If you have further concerns, please feel free to contact us.
> >
> > > **Q5**: Motivation of adversarial networks.
> >
> > **A5**: Medical image semantic segmentation can be formulated as a typical dense prediction problem, which aims at performing pixel-level classification on the feature maps. Despite recent advances in medical image segmentation, it still remains unclear whether it is sufficient to learn both low-level anatomical features and high-level semantics since all label variables are independently predicted from each other. To this end, the motivation of the adversarial network is to reinforce the spatial contiguity between the output label maps and the ground-truth segmentation maps by detecting and correcting the higher-order inconsistencies [2,34]. In this study, we propose a transformer-based discriminator. Such designs have the following benefits: (1) It enables the discriminator to model long-range dependencies, making it better assess the medical image fidelity; (2) it can prioritize the most informative demonstrations on interesting anatomical regions, and differentiate irrelevant regions (i.e., background) from the category label regions; (3) it essentially endows the model with a more holistic understanding of the anatomical visual modality (categorical features). Moreover, we further experimentally demonstrate the necessity of using the adversarial network compared to the model without the adversarial network (please see Table 3 Line 282 - 283). For your convenience, the following Table shows the comparison results of “w/ adversarial network” (CASTformer) and “w/o adversarial network” (CATformer) on the Synapse multi-organ CT dataset.
> >
> > |  Model  |  Dataset  |  DSC  |  Jaccard  |  95HD  |  ASD  |
> > |  :-----------  |  :-----------:  |  :-----------:  |  :-----------:  |  :-----------:  |  :-----------:  |
> > |  w/o adversarial network  |  Synapse  |  82.17  |  73.22  |  **16.20**  |  **4.28**  |
> > |  w/ adversarial network  |  Synapse  |  **82.55**  |  **74.69**  |  22.73  |  5.81  |
> > |  |  |  |  |  |  |
> > |  w/o adversarial network  |  LiTS  |  72.39  |  62.76  |  **22.38**  |  11.57  |
> > |  w/ adversarial network  |  LiTS  |  **73.82**  |  **64.91**  |  23.35  |  **10.16**  |
> > |  |  |  |  |  |  |
> > |  w/o adversarial network  |  MP-MRI  |  94.17  |  86.50  |  **6.55**  |  3.33  |
> > |  w/ adversarial network  |  MP-MRI  |  **94.93**  |  **87.81**  |  8.29  |  **3.02**  |
> >
> > As we can see, our CASTformer w/ adversarial networks can outperform that of w/o adversarial networks in terms of DSC and Jaccard, and achieve comparable performance in terms of 95HD and ASD. As shown in Figures 3 (Line 236 - 237), 4 (Line 258 - 259), and 5 (Line 677 - 678), our method is capable of predicting high-quality object segmentation, considering the fact that the improvements in such settings are challenging. This demonstrates: (1) the necessity of adaptively focusing on the region of interests; and (2) the efficacy of semantically correlated information. Moreover, we conduct a thorough analysis of different GAN-based loss functions in Appendix I.
> >
> > Overall, we greatly improved the clarity of the paper and employed a more appropriate set of experiments in this revision. We hope that the changes will make this work in better shape for publication. Please feel free to contact us for further concerns.

---

> ### Author Response · Authors · 2022-08-08
> **Look forward to further feedback**
>
> Dear Reviewer aCmr:
>
> Thanks again for all of your constructive suggestions, which have helped us improve the quality and clarity of the paper!
>
> Since the author-reviewer discussion period will end soon in 2 days, we appreciate it if you take the time to read our rebuttal and give us some feedback. Please don't hesitate to let us know if there are any additional clarifications or experiments that we can offer, as we would love to convince you of the merits of the paper. We appreciate your suggestions. If our response resolves your concerns, we kindly ask you to consider raising the rating of our work.
>
> Thanks for your time and efforts!
>
> Best,
>
> Authors of Paper4486

---

### Meta-Review · Area_Chair_wvvk · 2022-08-30

**Recommendation:** Accept
**Confidence:** Less certain

**Metareview:**

The paper proposes a generative adversarial approach to 2D medical image segmentation. The problem is a standard one for MRI and improvements in this direction can have real-world impact. The reviewers were on the whole positive in their opinions of the paper. They found the design to be well-motivated and the paper to be well-written and easy to follow. The improvement to current methods was significant enough to be a good reason for acceptance. The reviewers generally found the feedback period to be helpful in swaying them in a more positive direction for accepting the paper.

**Award:**

No

---

### Decision · Program_Chairs · 2022-09-14

Accept